# HTAP2 multi-model estimates of premature human mortality due to intercontinental transport of air pollution

Ciao-Kai Liang[1], J. Jason West[1], Raquel A. Silva[2], Huisheng Bian[3], Mian Chin[4], Yanko Davila[5], Frank J. Dentener[6], Louisa Emmons[7], Johannes Flemming[8], Gerd Folberth[9], Daven Henze[6], Ulas Im[10], Jan Eiof Jonson[11], Terry J. Keating[12], Tom Kucsera[13], Allen Lenzen[14], Meiyun Lin[15], Marianne Tronstad Lund[16], Xiaohua Pan[17], Rokjin J. Park[18], R. Bradley Pierce[19], Takashi Sekiya[20], Kengo Sudo[20], Toshihiko Takemura[21]

[1]Department of Environmental Sciences and Engineering, University of North Carolina at Chapel Hill, Chapel Hill, North Carolina, USA
[2]Oak Ridge Institute for Science and Education at US Environmental Protection Agency, Research Triangle Park, NC, USA
[3]Goddard Earth Sciences and Technology Center, University of Maryland, Baltimore, MD, USA
[4]Earth Sciences Division, NASA Goddard Space Flight Center, Greenbelt, MD, USA
[5]Department of Mechanical Engineering, University of Colorado, Boulder, CO, USA
[6]European Commission, Joint Research Center, Ispra, Italy
[7]Atmospheric Chemistry Observations and Modeling Laboratory, National Center for Atmospheric Research (NCAR), Boulder, CO, USA
[8]European Center for Medium-Range Weather Forecasts, Reading, UK
[9]UK Met Office Hadley Centre, Exeter, UK
[10]Aarhus University, Department of Environmental Science, Frederiksborgvej, DK-4000, Roskilde, Denmark.
[11]Norwegian Meteorological Institute, Oslo, Norway
[12]US Environmental Protection Agency, Research Triangle Park, NC, USA
[13]Universities Space Research Association, Greenbelt, MD, USA
[14]Space Science & Engineering Center, University of Wisconsin -Madison, WI, USA
[15]Atmospheric and Oceanic Sciences, Princeton University, Princeton, NJ, USA
[16]CICERO Center for International Climate Research, Oslo, Norway
[17]Earth System Science Interdisciplinary Center, University of Maryland, College Park, MD, USA
[18]Seoul National University, Seoul, Korea
[19]NOAA National Environmental Satellite, Data, and Information Service, Madison, WI, USA
[20]Nagoya University, Furocho, Chigusa-ku, Nagoya, Japan
[21]Research Institute for Applied Mechanics, Kyushu University, Fukuoka, Japan

Correspondence to: J. Jason West (jjwest@email.unc.edu)
**Abstract**
Ambient air pollution from ozone and fine particulate matter is associated with
premature mortality. As emissions from one continent influence air quality over others,
changes in emissions can also influence human health on other continents. We estimate
global air pollution-related premature mortality from exposure to $PM_{2.5}$ and ozone , and
the avoided deaths from 20% anthropogenic emission reductions from six source
regions, North America (NAM), Europe (EUR), South Asia (SAS), East Asia (EAS),
Russia/Belarus/Ukraine (RBU) and the Middle East (MDE), three global emission
sectors, Power and Industry (PIN), Ground Transportation (TRN) and Residential (RES)
and one global domain (GLO), using an ensemble of global chemical transport model
simulations coordinated by the second phase of the Task Force on Hemispheric
Transport of Air Pollution (TF-HTAP2), and epidemiologically-derived concentration-
response functions. We build on results from previous studies of the TF-HTAP by using
improved atmospheric models driven by new estimates of 2010 anthropogenic
emissions (excluding methane), with more source and receptor regions, new
consideration of source sector impacts, and new epidemiological mortality functions.
We estimate 290,000 (95% CI: 30,000, 600,000) premature $O_3$-related deaths and 2.8
million (0.5 million, 4.6 million) $PM_{2.5}$-related premature deaths globally for the
baseline year 2010. While 20% emission reductions from one region generally lead to
more avoided deaths within the source region than outside, reducing emissions from
MDE and RBU can avoid more $O_3$-related deaths outside of these regions than within,
and reducing MDE emissions also avoids more $PM_{2.5}$-related deaths outside of MDE
than within. Our findings that most avoided $O_3$-related deaths from emission reductions
in NAM and EUR occur outside of those regions contrast with those of previous studies,
while estimates of $PM_{2.5}$-related deaths from NAM, EUR, SAS and EAS emission
reductions agree well. In addition, EUR, MDE and RBU have more avoided $O_3$-related
deaths from reducing foreign emissions than from domestic reductions. For six regional
emission reductions, the total avoided extra-regional mortality is estimated as 6,000 (-
3,400, 15,500) deaths/year and 25,100 (8,200, 35,800) deaths/year through changes in
$O_3$ and $PM_{2.5}$, respectively. Interregional transport of air pollutants leads to more deaths
through changes in $PM_{2.5}$ than in $O_3$, even though $O_3$ is transported more on
interregional scales, since $PM_{2.5}$ has a stronger influence on mortality. For NAM and
EUR, our estimates of avoided mortality from regional and extra-regional emission
reductions are comparable to those estimated by regional models for these same
experiments. In sectoral emission reductions, TRN emissions account for the greatest

fraction (26-53% of global emission reduction) of $O_3$-related premature deaths in most regions, in agreement with previous studies, except for EAS (58%) and RBU (38%) where PIN emissions dominate. In contrast, PIN emission reductions have the greatest fraction (38-78% of global emission reduction) of $PM_{2.5}$-related deaths in most regions, except for SAS (45%) where RES emission dominates, which differs with previous studies in which RES emissions dominate global health impacts. The spread of air pollutant concentration changes across models contributes most to the overall uncertainty in estimated avoided deaths, highlighting the uncertainty in results based on a single model. Despite uncertainties, the health benefits of reduced intercontinental air pollution transport suggest that international cooperation may be desirable to mitigate pollution transported over long distances.

## 1 Introduction

Ozone ($O_3$) and fine particulate matter with aerodynamic diameter less than 2.5 μm ($PM_{2.5}$) are two common air pollutants with known adverse health effects. Epidemiological studies have shown that both short-term and long-term exposures to $O_3$ and $PM_{2.5}$ are associated with elevated rates of premature mortality. Short-term exposure to $O_3$ is associated with respiratory morbidity and mortality (Bell et al., 2005; Bell et al., 2014; Gryparis et al., 2004; Ito et al., 2005; Levy et al., 2005; Stieb et al., 2009) while long-term exposure to $O_3$ has been associated with premature respiratory mortality (Jerrett et al., 2009, Turner et al., 2016). Short-term exposure to $PM_{2.5}$ has been associated with increases in daily mortality rates from all natural causes, and specifically from respiratory and cardiovascular causes (Bell et al., 2014; Du et al., 2016; Powell et al., 2015; Pope et al., 2011) while long-term exposure to $PM_{2.5}$ can have detrimental chronic health effects, including premature mortality due to cardiopulmonary diseases and lung cancer (Brook et al., 2010; Burnett et al., 2014; Hamra et al., 2014; Krewski et al., 2009; Lepeule et al., 2012; Lim et al., 2012). The Global Burden of Disease Study 2015 (GBD 2015) estimated 254,000 deaths/year associated with ambient $O_3$ and 4.2 million associated with ambient $PM_{2.5}$ (Cohen et al. 2017). A comparable study using output from an ensemble of global chemistry–climate models estimated 470,000 deaths/year associated with $O_3$ and 2.1 million premature deaths/year associated with anthropogenic $PM_{2.5}$ (Silva et al. 2013). These differences in GBD estimates result mainly from differences in concentration response functions and estimates of pollutant concentrations.

Numerous observational and modeling studies have shown that anthropogenic emissions can affect $O_3$ and $PM_{2.5}$ concentrations across continents (Dentener et al., 2010; Heald et al., 2006; Leibensperger et al., 2011; Lin et al., 2012; Lin et al., 2017;

Liu et al., 2009a; West et al., 2009a; Wild and Akimoto, 2001; Yu et al., 2008). As changes in emissions from one continent influence air quality over others, several studies have estimated the premature mortality from intercontinental transport (Anenberg et al., 2009; Anenberg et al., 2014; Bhalla et al., 2014; Duncan et al., 2008; Im et al., 2018; Liu et al., 2009b; West et al., 2009b; Zhang et al., 2017). In 2005, the Task Force on Hemispheric Transport of Air Pollution (TF-HTAP) was launched under the United Nations Economic Commission for Europe (UNECE) Convention on Long-Range Transboundary Air Pollution (LRTAP). One of its tasks is to investigate the impacts of emission reductions on the intercontinental transport of air pollution, air quality, health, ecosystem and climate effects, using a multi-model ensemble to quantify uncertainties due to differences between models (Anenberg et al., 2009; Anenberg et al., 2014; Fiore et al., 2009; Fry et al., 2012; Huang et al., 2017; Stjern et al., 2016; Yu et al., 2013).

In the TF-HTAP Phase 1 (TF-HTAP1), human premature mortality due to 20% anthropogenic emission reductions in four large source regions was investigated by Anenberg et al. (2009 and 2014). They found that 20% foreign $O_3$ precursor emission reductions contribute approximately 30% to >50% of the deaths avoided by reducing precursor emissions in all four regions together (Anenberg et al., 2009). Similarly, reducing emissions in NA and EU was found to avoid more $O_3$-related premature deaths outside the source region than within (Anenberg et al., 2009), which agrees with other studies that together show for the first time that emission reductions in NA and EU have greater impacts on mortality outside the source region than within (Duncan et al., 2008; West et al., 2009). In contrast, Anenberg et al. (2014) estimate that 93–97 % of $PM_{2.5}$-related avoided deaths from reducing emissions in all four regions occurs within the source region while 3–7 % occur outside the source region from transport between continents. Despite the longer atmospheric lifetime of $O_3$ and its relatively larger scale of influence, $PM_{2.5}$ was found to cause more deaths from intercontinental transport (Anenberg et al., 2009; 2014). These prior studies have consistently concluded that most avoided $O_3$-related deaths from emission reductions in NAM and EUR occur outside of those regions, while most avoided $PM_{2.5}$-related deaths occur within the regions. Similarly, an ensemble of regional models in the third phase of the Air Quality Modelling Evaluation International Initiative (AQMEII3) found that a 20% decrease of emissions within the source region avoids 54,000 and 27,500 premature deaths in Europe and the U.S. (from both $O_3$ and $PM_{2.5}$), while the reduction of foreign emissions alone avoids ~1,000 and 2,000 premature deaths in Europe and the U.S. (Im et al., 2018). Crippa et al (2017) used the TM5-FASST reduced-form model with HTAP2 emissions to estimate a global sensitivity to 20 % emission reductions of $PM_{2.5}$-related premature deaths of 401,000 globally, and 42,000 and 20,000 for Europe and the US respectively.

In addition, several studies have evaluated the relative importance of individual
emissions sectors (Barrett et al., 2010; Bhalla et al., 2014; Chafe et al., 2014; Chambliss
et al., 2014; Corbett et al., 2007) or multiple sectors (Lelieveld et al., 2015; Silva et al.,
2016a) to ambient air pollution–related premature mortality. Lelieveld et al. (2015)
estimated that residential energy use such as for heating and cooking has the largest
mortality impact globally (for $PM_{2.5}$ and $O_3$ mortality combined), particularly in South
and East Asia. Silva et al (2016) likewise found that residential & commercial emissions
are most important for ambient $PM_{2.5}$-related mortality, but also found that land
transportation had the greatest impact on $O_3$-related mortality, particularly in North
America, South America, Europe, FSU and the Middle East. Understanding the impact
of different sectors on the global burden and the relative importance of each sector
among regions can help stimulate international efforts and region-specific air pollution
control strategies. Nevertheless, those studies were limited by using a single
atmospheric model, reflecting a need to understand whether results differ among
models and apportionment approaches.
In this study, we estimate the impacts of interregional transport and of source
sector emissions on human premature mortality from $O_3$ and $PM_{2.5}$, using an ensemble
of global chemical transport models coordinated by the Task Force on Hemispheric
Transport of Air Pollution Phase 2 (TF-HTAP2) (Galmarini et al., 2017; Huang et al.,
2017; Janssens-Maenhout et al., 2015; Stjern et al., 2016). Anthropogenic emissions
were reduced by 20% in six source regions: North America (NAM), Europe (EUR),
South Asia (SAS), East Asia (EAS), Russia/Belarus/Ukraine (RBU) and the Middle
East (MDE), three emission sectors: Power and Industry (PIN), Ground Transportation
(TRN) and Residential (RES), and one worldwide region (GLO). Human premature
mortality due to these reductions is calculated using a health impact function based on
a log-linear model for $O_3$ (Jerrett et al. 2009) and an integrated exposure-response
model for $PM_{2.5}$ (Burnett et al. 2014), within the six source regions and elsewhere in
the world. We conduct a Monte Carlo simulation to estimate the overall uncertainty due
to uncertainties in relative risk, air pollutant concentrations (given by the spread of
results among different models), and baseline mortality rates.

**2 Method**
**2.1 Modeled $O_3$ and $PM_{2.5}$ surface concentration**
Global numerical modelling experiments initiated by TF-HTAP2, the regional
experiments by the Air Quality Model Evaluation International Initiative (AQMEII)
over Europe and North America, and the Modelling Intercomparison Study-Asia
(MICS-Asia) were coordinated to perform consistent emission perturbation modelling

experiments across the global, hemispheric and continental/regional scales (Galmarini et al., 2017). Simulation periods, meteorology, emission inventories, boundary conditions, and model output are also consistent. The Joint Research Centre's (JRC) EDGAR (Emission Data Base for Global Research) team in collaboration with regional emission experts from the U.S. Environmental Protection Agency (US-EPA), European Monitoring and Evaluation Programme (EMEP), Centre on Emission Inventories and Projections (CEIP), Netherlands Organization for Applied Research (TNO), and the MICS-Asia Scientific Community and Regional Emission Activity Asia (REAS) provide a global emission inventory at $0.1^0$x$0.1^0$ resolution for TF-HTAP2 modeling experiments (Janssens-Maenhout et al., 2015). The emissions dataset was constructed for $SO_2$, $NO_X$, CO, NMVOC, $NH_3$, $PM_{10}$, $PM_{2.5}$, BC and OC and seven emission sectors (shipping, aircraft, land transportation, agriculture, residential, industry and energy) for the year 2010 (Fig. S1).

This study uses outputs from 14 global models / model versions (Table S1) participating in TF-HTAP2. Overall, TF-HTAP2 model resolutions are finer than in TF-HTAP1. In TF-HTAP2, each model performed a baseline simulation and sensitivity simulations where the anthropogenic emissions in a defined source region or sector were perturbed (reduced by 20% in most cases). Based on the number of models that simulated different experiments, we choose to focus on emission reductions from six source regions, three emission sectors, and one global domain. More specifically, all anthropogenic emissions are reduced by 20% in the North America (NAM), Europe (EUR), South Asia (SAS), East Asia (EAS), Russia/Belarus/Ukraine (RBU) and the Middle East (MDE) continental regions, in the Power and Industry (PIN), Ground Transportation (TRN) and Residential (RES) emission sectors globally, and in one global domain (GLO) (Fig. S2). Unlike TF-HTAP1 (Dentener et al., 2010) which defined rectangular regions that included ocean or some sparsely inhabited regions, TF-HTAP2 regions are defined by geopolitical boundaries.

We selected output from the models that provided temporally resolved volume mixing ratios of $O_3$ and mass mixing ratios of $PM_{2.5}$ ("mmrpm2p5") for the baseline and at least one regional or sectoral emission reduction scenario. Among the 14 models, 11 models reported $O_3$ and 8 reported $PM_{2.5}$ for regional emission perturbation scenarios, 4 models reported $O_3$ and 4 reported $PM_{2.5}$ for sectoral emission perturbation scenarios, and 10 models reported $O_3$ and 8 reported $PM_{2.5}$ for the global emission perturbation. All models used prescribed meteorology for the year 2010, although this meteorology was derived from different (re-)analysis products and not uniform across models. Modeled concentrations are processed by calculating metrics consistent with the underlying epidemiological studies to estimate premature mortality. For $O_3$, we calculate the average of daily 1-h maximum $O_3$ concentration for the 6 consecutive

months with the highest concentrations in each grid cell (Jerrett et al., 2009), for the baseline and each 20% emission reduction scenario. While some models reported hourly $O_3$ metrics, others only reported daily or monthly $O_3$. We include these models by first calculating the ratio of the 6-month average of daily 1-h maximum $O_3$ to the annual average of $O_3$ in individual grid cells, for models reporting hourly $O_3$, and then applying that ratio to the annual average of ozone for those models that only report daily or monthly $O_3$, following Silva et al. (2013; 2016b). For $PM_{2.5}$, we calculate the annual average $PM_{2.5}$ concentration in each cell using the monthly total $PM_{2.5}$ concentrations reported by each model ("mmrpm2p5"). Model results for these two metrics are then regridded from each model's native grid resolution (varying from $0.5°×0.5°$ to $2.8°×2.8°$) to a consistent $0.5°×0.5°$ resolution used in mortality estimation. We estimate regional and sectoral multi-model averages for each 20% emission reduction scenario in the year 2010, but for each perturbation case, we only include models that report both the baseline and perturbation cases.

## 2.2 Model evaluation

Measurements from multiple observation networks are employed in this study to evaluate the model performance around the world. We evaluate model performance for the 2010 baseline simulation for 11 TF-HTAP2 models for $O_3$ and 8 for $PM_{2.5}$ (Table S1). For $O_3$, we use ground level measurements from 2010 at 4,655 sites globally, collected by the Tropospheric Ozone Assessment Report (TOAR) (Schultz et al., 2017; Young et al., 2018). The TOAR dataset identifies stations as urban, rural and unclassified sites (Schultz et al., 2017). Model performance is evaluated for the average of daily 1-h maximum $O_3$ concentrations for the 3 consecutive months (3m1hmaxO$_3$) with the highest concentrations in each grid cell, including models that only report daily or monthly $O_3$ as described above. This metric for $O_3$ differs slightly from the 6-month average of daily 1-h maximum metric used for health impact assessment, and is chosen because TOAR reports the 3-month metric but not the 6-month metric. For $PM_{2.5}$, we compare the annual average $PM_{2.5}$, using $PM_{2.5}$ observations from 2010 at 3,157 sites globally selected for analysis by the Global Burden of Disease 2013 (GBD2013) (Forouzanfar et al., 2016). Statistical parameters including the normalized mean bias (NMB), normalized mean error (NME), and correlation coefficient (R) are selected to evaluate model performance.

Table S2 and S3 present statistical parameters of model evaluation for $O_3$ and $PM_{2.5}$, and Figures S3-S10 show the spatial $O_3$ and $PM_{2.5}$ evaluation as NMB around the world, and in North America, Europe and East Asia. For 3m1hmaxO$_3$, the model ensemble mean shows good agreement with measurements globally with NMB of 7.3% and NME of 13.2%, but moderate correlation with R of 0.53 (Table S2). For individual

models, 8 models (CAM-chem, CHASER_T42, CHASER_T106, EMEPrv48,
GEOSCHEMADJOINT, GEOS-Chem, GFDL_AM3 and HadGEM2-ES) overestimate
3m1hmax$O_3$ with NMB of 9.2% to 23% while 3 models (C-IFS, OsloCTM3.v2 and
RAQMS) underestimate by -10.8% to -19.4% globally (Figure S3). In the 6
perturbation regions, the model ensemble mean is also in good agreement with the
measurements, with -11.2% to 25.3% for NMB, 9.8% to 25.3% for NME, and -0.09 to
0.98 for R. The ranges of NMB for individual models are -18.1% to 32.3%, -24.1% to
21.3%, -24.5% to 45.0%, -26.4% to 24.5%, -30.5% to 20.3%, -35.3% to 5.4%, in NAM,
EUR, SAS, EAS, MDE, and RBU, respectively (Figure S4-S6). Note that some regions
(SAS, MDE, and RBU) have very few observations for model evaluation, making the
comparison less robust. The underestimated $O_3$ in the western US and overestimated
$O_3$ in eastern US in most models is very close to the model performance result of Huang
et al. (2017) who compare 8 TF-HTAP2 models with CASTNET observations (Figure
S4) , as well as earlier studies under HTAP1 (Fiore et al. 2009). Similarly, Dong et al.
(2018) find that $O_3$ is overestimated in EUR and EAS by 6 TF-HTAP2 models,
consistent with our ensemble mean result in these two regions (Figure S5-S6).
For $PM_{2.5}$, the model ensemble mean agrees well with measurements globally,
with NMB of -23.1%, NME of 35.4%, and R of 0.77 (Table S3). For individual models,
only 1 model (GEOSCHEMADJOINT) overpredicts $PM_{2.5}$ by 20.3%, while the other
7 models underpredict $PM_{2.5}$ by -60.9% to -7.4% around the world (Figure S7). In 6
perturbation regions, the model ensemble mean is also in good agreement with
measurements, with ranges of NMB of -49.7% to 19.4%, 21.2% to 49.7% for NME,
and 0.50 to 1.00 for R. The range of NMB for individual models are -46.6% to 13.9%,
-76.0% to 31.9%, -35.0% to 49.7%, -50.4% to 29.5%, -52.6% to 31.5%, and -74.1% to
-19.8%, in NAM, EUR, SAS, EAS, MDE, and RBU, respectively (Figure S8-S10).
Dong et al. (2018) shows that $PM_{2.5}$ is underestimated in EUR and EAS by 6 TF-HTAP2
models, consistent with our ensemble mean result in these two regions (Figure S9-S10).
Note that many observations used are located in urban areas, and models with coarse
resolution may not be expected to have good model performance. Also several models
neglect some $PM_{2.5}$ species, which may explain the tendency of models to
underestimate.

**2.3 Health impact assessment**
We use output from the TF-THAP2 model ensemble to estimate annual $O_3$- and
$PM_{2.5}$-related global cause-specific premature mortality and avoided mortality from the
20% regional and sectoral emission reductions, following the same methods used by
Silva et al. (2016a; 2016b). The annual $O_3$- and $PM_{2.5}$-related premature mortality is
calculated using a health impact function based on epidemiological relationships
between ambient air pollution concentration and mortality in each grid cell: $\Delta M =$
$y_0 \times AF \times Pop$, where $\Delta M$ is premature mortality, $y_0$ is the baseline mortality rate
(for the exposed population), $AF = 1 - 1/RR$ is the attributable fraction, where $RR$ is
relative risk of death attributable to the change in air pollutant concentration ($RR=1$
when there is no increased risk of death associated with a change in pollutant
concentration), and $Pop$ is the exposed population (adults aged 25 and older).
For $O_3$ mortality, we use a log-linear model for chronic respiratory mortality
(RESP) from the American Cancer Society (ACS) study (Jerrett et al 2009), following
recent studies including the GBD (Cohen et al., 2017), but Turner et al. (2016) recently
published new results for chronic ozone mortality, and adoption of these results would
lead to more ozone-related deaths overall (Malley et al., 2017). RR is calculated as:

$RR = e^{\beta \Delta x}$   (1)

where $\beta$ is the concentration-response factor, and $\Delta x$ corresponds to the change in
pollutant concentrations between simulations with perturbed emissions and the baseline
simulation. For $O_3$, RR = 1.040 (95% Confidence Interval, CI: 1.013-1.067 ) for a 10
ppb increase in $O_3$ concentrations (Jerrett et al., 2009), which from eq. 1 gives values
for $\beta$ of 0.00392 (0.00129-0.00649). We estimate $O_3$-related premature deaths due to
respiratory disease (RESP) based on decreases or increases in $O_3$ concentration (i.e. $\Delta x$)
due to 20% regional and sectoral emission reduction scenarios relative to the baseline.
For regional and sectoral reductions, we do not assume a low-concentration threshold
below which changes in $O_3$ have no mortality effects, as there is no clear evidence for
such a threshold, following Anenberg et al (2009; 2010) and Silva et al. (2013; 2016a,
b). However, we evaluate global $O_3$ premature mortality for the baseline 2010
simulation, relative to a counterfactual concentration of 37.6 ppb (Lim et al. 2012), for
consistency with GBD estimates (Cohen et al., 2017).
For $PM_{2.5}$ mortality, we apply the Integrated Exposure–Response (IER) model,
which is intended to better represent the risk of exposure to $PM_{2.5}$ at locations with high
ambient concentrations (Burnett et al., 2014). RR is calculated as:

For $z < z_{cf}$, $RR_{IER}(z) = 1$   (2)

For $z \geqq z_{cf}$, $RR_{IER}(z) = 1 + \alpha\{1 - exp[-\gamma(z - z_{cf})^\delta]\}$   (3)

where z is the $PM_{2.5}$ concentration in μg/m³ and $z_{cf}$ is the counterfactual concentration
below which no additional risk is assumed, and the parameters α, γ, and δ are used to
fit the function for cause-specific RR (Burnett et al., 2014). The overall $PM_{2.5}$-related
cause-specific premature deaths related to ischemic heart disease (IHD),
cerebrovascular disease (STROKE), chronic obstructive pulmonary disease (COPD)
and lung cancer (LC) are estimated using RRs per age group for IHD and STROKE and
RRs for all ages for COPD and LC. A uniform distribution from 5.8 μg/m³ to 8.8 μg/m³
is used for $z_{cf}$ as suggested by Burnett et al. (2014), which does not vary in space nor
time. For uncertainty analysis, we use results from 1,000 Monte Carlo simulations of
Burnett et al. (2014) to calculate RR in each grid cell by eq.2 or eq. 3. We estimate
avoided premature mortality in 20% emission perturbation experiments by taking the
difference in premature mortality estimates with the 2010 baseline. However, in the IER
model, the concentration–response function flattens off at higher $PM_{2.5}$ concentrations,
yielding different estimates of avoided premature mortality for identical changes in air
pollutant concentrations from less-polluted vs. highly-polluted regions. That is, one unit
reduction of air pollution may have a stronger effect on avoided mortality in regions
where pollution levels are lower (e.g., Europe, North America) compared with highly
polluted regions (e.g., East Asia, India), which would not be the case for a log-linear
function (Jerrett et al., 2009; Krewski et al., 2009). Therefore, using the IER model in
this study may result in smaller changes in avoided mortality in highly polluted areas
than using the linear model.
For the exposed population, we use the Oak Ridge National Laboratory's Landscan
2011 Global Population Dataset at approximately 1 km resolution (30"x30") (Bright et
al., 2012). For the population of adults aged 25 and older, we use ArcGIS 10.2
geoprocessing tools to estimate the population per 5-year age group in each cell by
multiplying the country level percentage in each age group by the population in each
cell. We obtained cause-specific baseline mortality rates for 187 countries from the
GBD 2010 mortality dataset (IHME, 2013). The population and baseline mortality per
age group were regridded to the 0.5°×0.5° grid (Table S4 and Fig. S11). Cause-specific
baseline mortality rates vary geographically, e.g. RESP and COPD are relatively more
dominant in South Asia, IHD in Europe, STROKE in Russia, and LC in North America.
Finally, we conduct 1,000 Monte Carlo simulations to propagate uncertainty from
baseline mortality rates, modeled air pollutant concentrations, and the RRs in health
impact functions. We use the reported 95% CIs for cause-specific baseline mortality
rates, assuming lognormal distributions. For modeled $O_3$ and $PM_{2.5}$ concentrations we
use the absolute value of the coefficient of variation among models in each grid cell,
for each 20% emission perturbation case minus the baseline, assuming a normal
distribution. For $O_3$ RRs, we use the reported 95% confidence intervals (CIs), assuming
a normal distribution. For $PM_{2.5}$ RRs, we use the parameter values (i.e. $\alpha$, $\gamma$, $\delta$ and $z_{cf}$)
of Burnett et al. (2014) for 1,000 simulations. One should acknowledge that the range
of modeled air pollution concentrations in an ensemble is not a true reflection of the
uncertainty in emissions to concentration relationships. The mean health outcome of
the 1,000 Monte Carlo simulations (the "empirical mean") may differ from the mean
when using the mean RR.
We also quantify the uncertainties in mortality due to the spread of air pollutant
concentrations across models, RRs, and baseline mortality rates, as contributors to the
overall uncertainty, expressed as a coefficient, of variation and compare the result with
the Monte-Carlo analysis estimate. To do so, we hold two variables at their mean values
and change the variable of interest within its uncertainty range; for example, using mean
RRs and baseline mortality rates, we analyze the spread of the model ensemble to
calculate the coefficient of variation caused by model uncertainty. Given that our
$0.5° \times 0.5°$ grid cell resolution can capture most of the population well in a given region,
uncertainty associated with population was assumed to be negligible. We estimate the
impacts of extra-regional emission reductions on mortality by using the Response to
Extra-Regional Emission Reduction (RERER) metric defined by TF-HTAP (Galmarini
et al., 2017):
$$RERER_i = \frac{R_{global} - R_{region,i}}{R_{global}} \quad (4)$$
where for a given region $i$, $R_{global}$ is the change in mortality in the global 20%
reduction simulation (GLO) relative to the base simulation, and $R_{region,i}$ is the change
in mortality in response to the 20% emission reduction from that same region $i$. A
RERER value near 1 indicates a strong relative influence of foreign emissions on
mortality within a region, while a value near 0 indicates a weak foreign influence. We
also estimate the total avoided extra-regional mortality from a source perspective as the
sum of avoided deaths outside of each of the 6 source regions, and from a receptor
perspective by summing $R_{global} - R_{region,i}$ for all 6 regions.

**3 Results**
**3.1 Response of O$_3$ and PM$_{2.5}$ concentrations to 20% regional and sectoral**
**emission reductions**
Previous TF-HTAP studies reported area-averaged concentrations to quantify
source-receptor relationships averaging concentrations over a region (Doherty et al.,
2013; Fiore et al., 2009; Fry et al., 2012; Huang et al., 2017; Stjern et al., 2016; Yu et
al., 2013). Here, we present the population-weighted concentration over a region, which
is more relevant for health. Among six receptor regions, the population-weighted multi-
model mean O$_3$ concentrations range from 48.38±8.05 ppb in EUR to 65.72±10.08 ppb
in SAS with a global average of 53.74±8.03 ppb, while the annual population-weighted
multi-model mean PM$_{2.5}$ concentrations range from 9.36±2.62 μg/m$^3$ in NAM to 39.27
±13.50 μg/m$^3$ in EAS with a global average of 25.98±5.05 μg/m$^3$ (Table 1 and S5-S6
and Figs.S12-S13).
For 20% perturbation scenarios, in general the impact on the multi-model mean
change in surface O$_3$ and PM$_{2.5}$ concentration is greater within the source region (i.e.,
domestic region) than outside of it (i.e., foreign region) (Figs. 1-2). This is also true for
individual model results (Figs. S14-S16). Among six source regions, the emission
reduction from SAS has the greatest impact on global population-weighted $O_3$
concentration (Tables 2 and S5), while that from EAS has greatest impact on $PM_{2.5}$
(Tables 3 and S6). The source-receptor pairs with the greatest changes in $O_3$ and $PM_{2.5}$
concentration reflect the geographical proximity between regions and the magnitude of
emissions (Table 2-3) – e.g., EUR→MDE (0.34±0.08 ppb), EUR→RBU (0.34
ppb±0.09), EAS→NAM (0.29±0.14 ppb), EAS→RBU (0.27±0.12 ppb), and
NAM→EUR (0.26±0.55 ppb) for $O_3$, and EUR→RBU (0.26±0.19 $\mu g/m^3$), EUR→MDE
(0.18±0.08 $\mu g/m^3$), MDE→SAS (0.12±0.06 $\mu g/m^3$), SAS→EAS (0.08±0.08 $\mu g/m^3$),
and EAS→SAS (0.08±0.07 $\mu g/m^3$) for $PM_{2.5}$. Our ensemble shows similar ozone
responses in the western US to emission reductions from EAS (Figs. 1c) as those
modeled by Lin et al. (2012 and 2017), who show that a model can capture the measured
western US ozone increases due to rising Asian emissions.
For each receptor region, reducing foreign anthropogenic emissions by 20%
(estimated by global minus within-region reductions) can decrease population-
weighted $O_3$ concentrations by 29-74% of the change in $O_3$ concentration and 8–41 %
of the change in $PM_{2.5}$ concentration (Tables 2-3). In some cases, regional emission
reductions cause small $O_3$ concentration increases within the source region or in foreign
receptors, reflecting $O_3$ nonlinear responses (Figs. S14). For instance, C-IFS_v2
predicts $O_3$ concentration increases in EUR by 0.04 ppb from domestic emission
reductions, which is in agreement with results from TF-HTAP1 (Anenberg et al. 2009).
Similarly, CMAchem shows more local $O_3$ increases, particularly in SAS, than other
models (Figs. S14). The change in $O_3$ concentration in foreign receptors is broader than
for $PM_{2.5}$, reflecting that $O_3$ has a longer atmospheric lifetime than $PM_{2.5}$.
For sectors, TRN emission reductions cause the greatest decrease in global
population-weighted $O_3$ by 1.13±0.19 ppb, while PIN emission reductions cause the
greatest decrease in surface $PM_{2.5}$ by 1.46±0.56 $\mu g/m^3$ globally (Tables 2-3 and Figs. 1-
2). The 20% emission reductions from individual sectors also have different effects in
different regions. Of the three sectors, emission reductions from TRN have the greatest
effect on population-weighted $O_3$ in NAM, EUR, SAS, MDE and MDE (40-50% of the
global emission reduction) while PIN emission reductions dominate in EAS (57%).
Emission reductions from PIN have the greatest effect on population-weighted $PM_{2.5}$
in NAM, EUR, EAS, MDE and MDE (41-84%) while RES emission reductions
dominate in SAS (43%). The response of $PM_{2.5}$ concentration to sectoral emission
reductions differs significantly across models, which reflects in part the $PM_{2.5}$ species
simulated by each model (Table S1 and Figs. S15-S17). For instance, we found that
models that simulate $PM_{2.5}$ nitrate (i.e. CHASER_t42 and GEOSCHEMADJOIN)
predict a greater impact on PM$_{2.5}$ concentration from TRN emission reduction than
those without nitrate (i.e. GOCARTv5 and SPRINTARS) (Fig. S17).

**3.2 Global mortality burden associated with anthropogenic air pollution**
Table 4 shows the annual multi-model mean O$_3$- and PM$_{2.5}$-related premature
deaths on 6 regions and globally for year 2010 baseline with 95% confidence intervals
(CI) based on Monte Carlo sampling. Tables S7-S8 show estimates of premature deaths
due to anthropogenic O$_3$ and PM$_{2.5}$ from individual models. For the ensemble model
mean, we estimate 290,000 (30,000, 600,000) premature O$_3$-related deaths globally
using a 37.6 ppb counterfactual concentration, and 2.8 million (0.5 million, 4.6 million)
PM$_{2.5}$-related premature deaths using a uniform distribution of counterfactual
concentration from 5.8 μg/m$^3$ to 8.8 μg/m$^3$. Highly populated areas of India and East
Asia have the greatest O$_3$- and PM$_{2.5}$-related deaths, and those regions together account
for 82% and 66% of the global total O$_3$- and PM$_{2.5}$-related deaths. Compared with the
GBD 2015 (Cohen et al 2017), our global burden estimates are greater than the 254,000
(97,000, 422,000) premature deaths/year for O$_3$ from GBD, while less than 4.2 million
(3.7 million, 4.8 million) premature deaths for PM$_{2.5}$. Lelieveld et al (2015) estimate
142,000 (CI: 90,000, 208,000) O$_3$-related deaths and 3.2 million (1.5 million, 4.6
million) PM$_{2.5}$-related premature deaths for 2015. These differences can be explained
mainly by exposure estimates. Here we used a multi-model ensemble, whereas
Lelieveld et al. (2015) used a single model, and Cohen et al (2017) used a single model
for O$_3$ and a single model combined with surface and satellite observations for PM$_{2.5}$.
In addition, Cohen et al. (2017) use RRs for particulate matter for IHD and stroke
mortality that are modified from those used by Burnett et al (2014) and applied age
modification to the RRs, fitting the IER model for each age group separately. The
updated IER with estimated higher relative risks, together with greater global pollution
and baseline mortality rates in the low-income and middle-income countries in east and
south Asia leads to the higher absolute numbers of attributable deaths and disability-
adjusted life-years in GBD 2015 than estimated in GBD 2013 (Forouzanfar et al., 2016).
Also, GBD 2015 includes child lower respiratory infections estimate whereas we do
not. Our wider range of uncertainty for the global mortality reflects the uncertainty in
baseline rates, RRs and spread of air pollutant concentration across models whereas
Cohen et al (2017) consider national-level population-weighted mean concentrations
and uncertainty of IER function predictions at each concentration and Lelieveld et al.
(2015) only account for the statistical uncertainty of the parameters used in the IER
functions.

**3.3 Effect of regional reductions on mortality**

Reducing global anthropogenic emissions of air pollutant by 20% avoids 47,400 (11,300, 99,000) $O_3$-related deaths and 290,000 (67,100, 405,000) $PM_{2.5}$-related premature deaths (Tables 5-6 and S9-S10). Most avoided air pollution-related deaths were found within or close to the source region (Figs.3-76). Reducing anthropogenic emissions by 20% from NAM, EUR, SAS, EAS, MDE and RBU can avoid 54%, 54%, 95%, 85%, 21%, and 22% of the global change in $O_3$-related deaths within the source region (The number of avoided deaths within source region is divided by the number of avoided deaths globally), and 93%, 81%, 93%, 94%, 32%, and 82% of the global change in $PM_{2.5}$-related deaths, respectively (Table 5-6). Whereas the most $O_3$-related premature deaths can be avoided by reducing SAS emissions (20,000 (3,600, 42,200) deaths/year), reducing EAS emissions avoids more $O_3$-related premature deaths (1,700 (-1,300, 5,400)) outside of the source region than for any other region (500 (180, 870) deaths/year to 1,300 (-1,200, 4,400) deaths/year (Table 5). Similarly, while reducing EAS emissions avoids the most $PM_{2.5}$-related premature deaths (96,600 (3,500, 136,000) deaths/year), reducing EUR emissions avoids more $PM_{2.5}$-related premature deaths (7,400 (930, 9,500) deaths/year) outside of the source region than for any other region (1,400 (-320, 2,300) deaths/year to 5,500 (3,000, 7,800) deaths/year) (Table 6). While emission reductions from one region generally lead to more avoided deaths within the source region than outside, 20% anthropogenic emission reductions from MDE (i.e. 79% and 68% of global avoided deaths outside of source region for $O_3$ and $PM_{2.5}$, respectively) and RBU (78% for $O_3$) can avoid more premature deaths outside of the source region than within (Table 5-6). This result for RBU is in agreement with West et al (2009). However, the results for NAM and EUR do not agree with previous studies that found that emission reductions in these regions cause more $O_3$-related avoided premature deaths outside of the source region than within (Anenberg et al., 2009; Duncan et al., 2008; West et al., 2009). For $PM_{2.5}$, our results are comparable with Anenberg et al. (2014) and Crippa et al. (2017) who found that for most regions, $PM_{2.5}$-related avoided premature deaths are higher within the source region than outside. The above difference in results with TF-HTAP1 may be in part because of the definition of regions. Whereas the TF-HTAP2 regions are defined by geopolitical boundaries, the TF-HTAP1 regions are defined by square domains which are larger and include more ocean areas (Anenberg et al., 2009). In addition, updated atmospheric models and emissions inputs, as well as different atmospheric dynamics in the single years chosen in TF-HTAP1 vs. TF-HTAP2 may contribute to the differences.

Using individual models, different conclusions may result for the relative importance of inter-regional transport. For example, for $O_3$, 8 models predict that NAM emission reductions cause more $O_3$-related premature deaths within NAM (i.e CAM-

Chem, CHASER_T42, CHASER_T106, C-IFS, GEOSCHEMADJOINT, GEOS-
Chem, GFDL_AM3 and HadGEM2-ES), whereas 2 models predict more deaths outside
(i.e. EMEPrv48 and OsloCTM3.v2). 5 models suggest that EUR emission reductions
cause more $O_3$-related premature deaths within EUR (i.e. CAM-chem, CHASER_T42,
CHASER_T106, GFDL_AM3 and HadGEM2-ES), whereas 4 show more deaths
outside (i.e. C-IFS, GEOSCHEMADJOINT, EMEPrv48 and OsloCTM3.v2). Each
individual model shows that emission reductions from SAS and EAS avoid more $O_3$-
related premature deaths within than outside, and that those from MDE and RBU avoid
more $O_3$-related premature deaths outside than within (Fig. S18). For $PM_{2.5}$, each
individual model shows that emission reductions from NAM, EUR, SAS, EAS and
RBU avoid more $PM_{2.5}$-related premature deaths within than outside, while for
emission reductions from MDE, 3 models (EMEPrv48, GEOSCHEMADJOINT and
SPRINARS) show more $PM_{2.5}$-related premature deaths within, while 3
(CHASER_T42 GEOS5 and GOCART) show more $PM_{2.5}$-related premature deaths
outside (Fig. S19). The variation of health effect reflects the differences in processing
of natural emissions, atmospheric physical and chemical mechanisms, numerics etc
across models.
For each receptor region, reducing domestic anthropogenic emissions by 20%
contributes about 66%, 39%, 84%, 72%, 45% and 25% of the total $O_3$-related avoided
premature mortality (from the global reduction), and 90%, 78%, 87%, 87%, 58% and
66% of the total $PM_{2.5}$-related avoided premature mortality (from the global reduction)
in NAM, EUR, SAS, EAS, MDE and RBU, respectively (Table 5-6). Therefore,
reducing emissions from foreign regions avoids more $O_3$ premature deaths in EUR
(foreign emission account for 61% of total avoided deaths from the global reduction),
MDE (55%) and RBU (75%) than reducing domestic emissions (Table 5-6), in
agreement with the results for EUR from Anenberg et al (2009). Whereas EAS has the
greatest number of avoided $O_3$-related premature deaths due to foreign emission
reduction (3,800 (3,600, 3,900) deaths/year), RBU has the greatest fraction of $O_3$
mortality from foreign emission reductions (75%) (Table 5). Similarly, for $PM_{2.5}$, while
EAS has greatest number of avoided $PM_{2.5}$-related premature deaths due to foreign
emission reductions (13,600 (3,500, 18,800) deaths/year), MDE has the greatest
fraction of $PM_{2.5}$ mortality from foreign emission reduction (42%) (Table 6).
Overall, adding results from all 6 regional reductions, interregional transport of air
pollution from extra-regional contributions is estimated to lead to more avoided deaths
through changes in $PM_{2.5}$ (25,100 (8,200, 35,800) deaths/year) than in $O_3$ (6,000 (-3,400,
15,500) deaths/year), consistent with Anenberg et al. (2009; 2014). This result is due to
the greater influence of $PM_{2.5}$ on mortality, despite the shorter atmospheric lifetime of
$PM_{2.5}$ relative to $O_3$.
The contributions of different factors to the overall uncertainties in mortality are
shown in Tables S11-S12, considering uncertainties due to the spread of air pollutant
concentrations across models, RRs, and baseline mortality rates, expressed as
coefficients of variation. For both $O_3$ and $PM_{2.5}$ mortality, the spread of model results
generally contributes most to the overall uncertainty, followed by uncertainty in RRs
and in baseline mortality rates, for most source-receptor pairs. The spread of model
results is generally wider for $PM_{2.5}$ (14% to 3974% among source-receptor pairs) than
for $O_3$ (13% to 1065%). The uncertainty in RRs for $O_3$ mortality has constant value
(33% to 34%) due to the fixed uncertainty range of RRs from Jerrett et al. (2009),
whereas $PM_{2.5}$ mortality leads to a wider range of uncertainty (1% to 247%) in RRs
because the uncertainty differs at different $PM_{2.5}$ concentrations (Burnett et al., 2014).
Low uncertainty in baseline mortality rate was found for most source-receptor pairs
(<20%) except for the response of $PM_{2.5}$ mortality in SAS to 20% reduction from RBU
(66%).

**3.4 Effect of sectoral reductions on mortality**
Reducing global anthropogenic emissions by 20% in 3 sectors (i.e. PIN, TRN and
RES) together avoids 48,500 (7,100, 108,000) $O_3$-related premature deaths and 243,000
(66,800, 357,000) $PM_{2.5}$-related premature deaths globally (Tables 5-6), with the
greatest avoided air pollution-related premature deaths located in highly populated
areas (e.g., North America, Europe, India, China, etc.) (Figs.3-6). For instance, reducing
anthropogenic emissions by 20% in 3 sectors together avoids the highest number of $O_3$-
related deaths in SAS (24,000 (6,000, 49,600) deaths/year) and $PM_{2.5}$-related deaths in
EAS (83,400 (29,400, 135,000) deaths/year). We compare our estimates of $O_3$ and
$PM_{2.5}$-related premature deaths attributable to PIN, TRN and RES emissions with
previous studies, by multiplying our results for 20% emission reductions by 5, and by
combining their sectors to nearly match each of the three sectors in this study (Table 7).
Compared with Silva et al (2016a), our estimate of $O_3$ and $PM_{2.5}$-related premature
deaths attributable to PIN and TRN are very comparable, but that to RES is lower here.
In comparison with Lelieveld et al (2015), we estimate greater $O_3$ and $PM_{2.5}$-related
premature deaths attributable to PIN and TRN, but less for RES.
Like Silva et al. (2016a) and Lelieveld et al. (2015), different locations show
relatively different mortality responses to changes in sectoral emissions. Whereas PIN
emission reductions cause the greatest number of avoided $O_3$-related premature deaths
globally (19,300 (1,400, 45,000) deaths/year), TRN emission reductions cause the
greatest fraction of avoided deaths in most of the six regions (26-53% of the global
emission reduction), except for EAS (58%) and RBU (38%) where the effect of
reducing PIN emissions dominates. In comparison with other studies (Table 7), our
conclusion that PIN emissions cause the most $O_3$-related deaths and TRN emissions
cause the greatest fraction of avoided deaths in most regions agrees well with Silva et
al (2016a). For $PM_{2.5}$, reducing PIN emissions avoids the most $PM_{2.5}$-related premature
deaths globally (128,000 (41,600, 179,000) deaths/year) and in most regions (38-78%
of the global emission reduction), except for SAS (45%) where the RES emission
dominates. Although these findings differ from those of Lelieveld et al (2015) and Silva
et al (2016), who find that Residential emissions have the greatest of impact on $PM_{2.5}$
mortality globally and in most regions, all studies agree that PIN emissions have the
greatest impact in NAM. Our result is also comparable with Crippa et al (2017) who
find that PIN emissions have the greatest health impact in most countries. Although
comparable emission inventories are used (i.e. Lelieveld et al (2015) and this study use
EDGAR emissions while Silva et al (2016) use RCP8.5 emissions), our lower mortality
estimate for RES emissions may be explained by our 20% reductions relative to the
zero-out method, and the different years simulated.

Considering results from individual models, we found that mortality from TRN
emission reductions show greater relative uncertainty than from PIN or RES (Table 5-
6 and Table S9-S10), reflecting a greater spread of results across models. Regional
impacts from individual model also differ from the ensemble mean result - e.g., for $O_3$,
GEOSCHEMADJOINT and OsloCTM3.v2 show that reducing PIN emissions causes
the greatest fraction of avoided $O_3$-related deaths in EUR, while
GEOSCHEMADJOINT, HadGM2-ES and OsloCTM3.v2 show that TRN emissions
have the greatest fraction of avoided $O_3$-related deaths in RBU (Figs. S20). For $PM_{2.5}$,
CHASER_t42 and GEOSCHEMADJOINT show that reducing PIN emissions causes
the greatest fraction of avoided $PM_{2.5}$-related deaths in SAS (Figs. S21).

**4 Discussion**

We aggregate the avoided deaths attributable to 20% reductions from four
corresponding source regions (i.e. NAM, EUR, SAS and EAS), and compare with the
findings from TF-HTAP1. We estimate that these regional emission reductions are
associated with 36,000 (-1,500, 90,300) avoided deaths globally through the change in
$O_3$ and 207,000 (41,500, 304,000) avoided deaths through the change in $PM_{2.5}$, more
than those estimated by Anenberg et al. (2009 and 2014) – 21,800 (10,600, 33,400)
deaths for $O_3$ and 192,000 (146,000, 230,000) deaths for $PM_{2.5}$. This discrepancy might
be attributed to different health impact function, emissions data sets, region definitions,
updated population or baseline mortality rates. In particular, for $O_3$ respiratory mortality,
we use a log-linear model for chronic mortality (Jerrett et al 2009), instead of the short-
term $O_3$ mortality estimate based on a daily time-series study (Bell et al., 2004) used by

Anenberg et al., (2009). For $PM_{2.5}$ mortality, Anenberg et al., (2014) only included the simulated changes in BC, particulate organic matter (POM=primary organic aerosol+secondary organic aerosol), and sulfate for $PM_{2.5}$ concentration, while we use the total model reported $PM_{2.5}$ concentration which includes more species for some models. We also apply the Integrated Exposure–Response (IER) model (Burnett et al. 2014) for $PM_{2.5}$, as opposed to the log-linear model of Krewski et al. (2009) used by Anenberg et al., (2014).

For regional reductions, our multi-model average results suggest that NAM and EUR emissions cause more deaths inside of those regions than outside, which disagrees with previous studies (Anenberg et al., 2009; Duncan et al., 2008; West et al., 2009) whereas similar regional impacts are found for EAS and SAS. Also, total avoided deaths through interregional air pollution transport are estimated as 6,000 (-3,400, 15,500) deaths/year for $O_3$ and 25,100 (8,200, 35,800) deaths/year for $PM_{2.5}$ in this study, in contrast with 7,300 (3,600, 11,200) deaths/year for $O_3$ and 11,500 (8,800, 14,200) deaths/year for $PM_{2.5}$ in Anenberg et al. (2009; 2014). These differences likely result from different concentration-response functions and the use of 6 regions here vs. 4 by Anenberg et al. (2009; 2014). In addition, updated atmospheric models and emissions inputs, as well as different atmospheric dynamics in the single years chosen in TF-HTAP1 vs. TF-HTAP2 may contribute to the differences. In addition, updated atmospheric models and emissions inputs, as well as different atmospheric dynamics in the single years chosen in HTAP vs. HTAP2 may contribute to the differences.   Overall, whereas $O_3$ accounts for a higher percentage of the total deaths in foreign regions than $PM_{2.5}$, $PM_{2.5}$ leads to more deaths in general, which agrees well with the results of Anenberg et al. (2009; 2014).

Using regional models in AQMEII3, driven by a single global model (C-IFS_v2), Im et al. (2018) estimated that 20% domestic emission reductions would avoid 54,000 and 27,500 premature deaths (for $O_3$ and $PM_{2.5}$ combined) in Europe and the U.S., respectively, as opposed to ~1,000 and 2,000 premature deaths due to foreign emission reductions. These results are comparable to our estimates that 32,900 and 19,500 premature deaths result from 20% domestic emission reductions in Europe and the U.S., while 670 and 570 premature deaths result from foreign emission reductions. Although our defined U.S. region is slightly bigger than Im et al. (2018), the majority of U.S. emission sources and population are located within the region defined by Im et al. (2018). This comparison shows that regional and global models show similar impacts on mortality from air pollution transport.

Differences in our estimates of premature mortality attributable to air pollution from three emission sectors (multiplied by 5) may be explained by methodological differences relative to previous studies (Silva et al., 2016; Lelieveld et al., 2015),

including our use of 20% emission reductions versus the zero-out method in those
studies, different emission inventories, a multi-model ensemble versus single models,
and differences in baseline mortality rates, population, and concentration response
functions. Our finding that TRN emissions contribute the most avoided deaths for $O_3$
in most regions agrees well with the result by Silva et al (2016a), but differs for $PM_{2.5}$
mortality for which we find that PIN emissions cause the most deaths, while both Silva
et al (2016a) and Lelieveld et al (2015) find that RES emissions are responsible for the
most deaths. This discrepancy may be explained by different $PM_{2.5}$ species included in
individual models, as we showed that changes in $PM_{2.5}$ concentration to TRN emission
differ across models.

By using an ensemble of multi-model results here, we highlight the relative
importance of difference source-receptor pairs for mortality in a way that is more robust
than using a single model, particularly since some individual models yielded different
conclusions than the ensemble mean. The air pollutant concentration changes reported
by the HTAP2 models may be different among models, it may result from variety of
processes, e.g. atmospheric physical and chemical mechanisms, processing of natural
emissions, and transport time step, etc. (Table S1), but not anthropogenic emissions
since those were nearly identical among models. In addition, the coarse model
resolution used by global models may underestimate health effects by misaligning peak
concentration and population, particularly in urban areas and for $PM_{2.5}$ (Punger and
West, 2013), but it is not known how model resolution would affect the relative
contributions of extra-regional and intraregional health benefits. Future research should
explore the possible bias from using coarse global models for extra-regional and
intraregional mortality estimates in metropolitan regions by comparing with finer-
resolution chemical transport models.

Another uncertainty in this paper (and other global studies) lies in applying the
same RRs worldwide, because of lack of long-term records of the chronic influences of
ambient air pollution on mortality outside of North America and Europe. We consider
only the population of adults ≥25 years old, ignoring possible mortality effects on the
younger population, and consequently we may underestimate premature mortality
overall. Likewise, the effects of air pollution on several morbidity endpoints are omitted.
We assume that all $PM_{2.5}$ is equally toxic, for lack of clear evidence for greater toxicity
of some species. Inter-regional transport may also change the toxicity of $PM_{2.5}$ by
changing the size distribution or chemical composition, where transport likely causes
particles to become more oxidized (West et al., 2016). Future research on $PM_{2.5}$-related
mortality should include estimating health effects for different $PM_{2.5}$ chemical
components.

## 5 Conclusions

We estimate $O_3$- and $PM_{2.5}$-related premature mortality from simulations with 14 global CTMs participating in the TF-HTAP2 multi-model exercise for the year 2010. An estimate of 290,000 (30,000, 600,000) global premature $O_3$-related deaths and 2.8 million (0.5 million, 4.6 million) global $PM_{2.5}$-related premature deaths is obtained from the ensemble for the year 2010 in the baseline case. We focus on model experiments simulating 20% regional air pollutant emission reductions (excluding methane) in 6 regions, 3 sectors and 1 global domain. For regional scenarios, 6 source emission reductions altogether can cause 84% of the global avoided $O_3$-related premature deaths within the source region, ranging from 21 to 95% among 6 regions, and 16% (5 to 79%) outside of the source region. For $PM_{2.5}$, 89% of global avoided $PM_{2.5}$-related premature deaths are within the source region, ranging from 32 to 94% among 6 regions, and 11% (6 to 68%) outside of the source region. While most avoided mortality generally occurs within the source region, we find that emission reductions from RBU (only for $O_3$) and MDE (for both $O_3$ and $PM_{2.5}$) can avoid more premature deaths outside of these regions than within. Considering the effects of foreign emissions on receptor regions, 20% foreign emission reductions lead to more avoided $O_3$-related premature deaths in EUR, MDE and RBU than domestic reductions. Reductions from all six regions in the transport of air pollution between regions are estimated to lead to more avoided deaths through changes in $PM_{2.5}$ (25,100 (8,200, 35,800) deaths/year) than for $O_3$ (6,000 (-3,400, 15,500) deaths/year). For NAM and EUR, our estimates of avoided mortality from regional and extra-regional emission reductions are comparable to those estimated by regional models in AQMEII3 (Im et al., 2018) for these same emission reduction experiments. Overall, the spread of modeled air pollutant concentrations contributes most to the uncertainty in mortality estimates, highlighting that using a single model may lead to erroneous conclusions and may underestimate uncertainty in mortality estimates.

For sectoral emission reductions, reducing anthropogenic emissions by 20% in 3 sectors together avoids 48,500 (7,100, 108,000) $O_3$-related premature deaths and 243,000 (66,800, 357,000) $PM_{2.5}$-related premature deaths globally. Of the 3 sectors, TRN had the greatest fraction (26-53%) of $O_3$-related premature deaths globally and in most regions, except for EAS (58%) and RBU (38%) where PIN emissions dominate. For $PM_{2.5}$ mortality, PIN emissions cause the most deaths in most regions (38-78%), except for SAS (45%) where the TRN emissions dominate.

In this study, we have gone beyond previous TF-HTAP1 studies that quantified premature mortality from interregional air pollution transport, by using more source regions, analyzing source emission sectors, and using updated atmospheric models and

health impact functions. The estimate of air transport premature mortality could vary
due to differences in exposure estimate (single model vs ensemble model), health
impact function, regional definitions, and grid resolutions. These discrepancies
highlight uncertainty estimated by different methods in previous studies. Despite
uncertainties, our results suggest that reducing pollution transported over a long
distance would be beneficial for health, with impacts from all foreign emission
reductions combined that may be comparable to or even exceed the impacts of emission
reductions within a region. Additionally, actions to reduce emissions should target
specific sectors within world regions, as different sectors dominate the health effects in
different regions. This work highlights the importance of long-range air pollution
transport, and suggests that estimates of the health benefits of emission reductions on
local, national, or continental scales may underestimate the overall health benefits
globally, when interregional transport is accounted for. International cooperation to
reduce air pollution transported over long distances may therefore be desirable.

**Acknowledgments** We sincerely acknowledge the contribution of modeling
groups from the second phase of Task Force on Hemispheric Transport of Air Pollution
(TF-HTAP2). This work was supported by a scholarship from the Taiwan Ministry of
Education, grants from NIEHS (1 R21 ES022600-01), and NASA (NNX16AQ30G and
NNX16AQ26G), funding from BEIS under the Hadley Centre Climate Programme
contract (GA01101) and from the European Union's Horizon 2020 research and
innovation programme under grant agreement no. 641816 (CRESCENDO). The
National Center for Atmospheric Research is sponsored by the National Science
Foundation. We thank Dr. Owen Cooper who provided the TOAR ground level $O_3$
observation dataset, and Michael Brauer for the GBD2013 ground level $PM_{2.5}$
observation dataset.

**Supporting information** A detailed description of the models participating in the
ensemble, a map of six priority regions used in this analysis, and additional results can
be found in the Supporting Information.

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

Table 1. Population-weighted multi-model mean $O_3$ (ppb) and $PM_{2.5}$ concentration ($\mu g/m^3$) for the 2010 baseline, for the 6-month $O_3$ season
average of 1-hr. daily maximum $O_3$ and annual average $PM_{2.5}$, shown with the standard deviation among models.

| Scenarios | Receptor regions | | | | | | |
|---|---|---|---|---|---|---|---|
| | NAM | EUR | SAS | EAS | MDE | RBU | World |
| $O_3$ (11 models) | 56.51±9.40 | 48.38±8.05 | 65.72±10.08 | 59.10±10.46 | 61.11±9.79 | 46.79±7.53 | 53.74±8.03 |
| $PM_{2.5}$ (8 models) | 9.36±2.62 | 10.75±3.87 | 37.05±8.74 | 39.27±13.50 | 34.49±17.64 | 11.61±3.52 | 25.98±5.05 |











Table 2. Population-weighted multi-model mean change in $O_3$ (ppb) in receptor regions due to 20% regional (NAM, EUR, SAS, MDE and RBU),
sectoral (PIN, TRN and RES) and global (GLO) anthropogenic emission reductions, for the 6-month $O_3$ season average of 1-hr. daily maximum.
The diagonal, showing the effect of each region on itself, is underlined. All numbers are rounded to the nearest hundredth, and are shown with
standard deviations among models.

| Source | Receptor region | | | | | | |
|--------|------|------|------|------|------|------|------|
| regions/sectors | NAM | EUR | SAS | EAS | MDE | RBU | World |
| NAM | -1.88±0.06 | -0.26±0.55 | -0.04±0.14 | -0.11±0.06 | -0.23±0.12 | -0.21±0.09 | -0.19±0.07 |
| EUR | -0.08±0.04 | -0.80±0.55 | 0.01±0.14 | -0.10±0.06 | -0.34±0.08 | -0.34±0.09 | -0.14±0.07 |
| SAS | -0.05±0.02 | -0.04±0.02 | -3.65±0.94 | -0.08±0.04 | -0.11±0.04 | -0.04±0.03 | -0.90±0.22 |
| EAS | -0.29±0.14 | -0.25±0.13 | -0.09±0.22 | -1.96±1.10 | -0.23±0.12 | -0.27±0.12 | -0.58±0.25 |
| MDE | -0.04±0.02 | -0.05±0.01 | -0.07±0.15 | -0.03±0.01 | -1.23±0.66 | -0.11±0.01 | -0.09±0.04 |
| RBU | -0.05±0.04 | -0.13±0.05 | 0.03±0.16 | -0.08±0.06 | -0.10±0.07 | -0.45±0.38 | -0.05±0.06 |
| PIN | -1.13±0.28 | -0.70±0.19 | -1.43±0.18 | -1.58±0.88 | -1.09±0.45 | -0.69±0.31 | -1.11±0.25 |
| TRN | -1.26±0.42 | -0.81±0.34 | -2.05±0.32 | -0.73±0.32 | -1.40±0.17 | -0.71±0.19 | -1.13±0.19 |
| RES | -0.24±0.09 | -0.21±0.04 | -1.19±0.44 | -0.62±0.10 | -0.23±0.06 | -0.18±0.03 | -0.57±0.14 |
| GLO | -2.86±0.77 | -1.98±0.66 | -4.40±1.04 | -2.77±1.21 | -2.84±0.70 | -1.76±0.52 | -2.82±0.53 |


Table 3. Population-weighted multi-model annual average change in PM$_{2.5}$ concentrations ($\mu$g/m$^3$) in receptor regions due to 20% regional (NAM,
EUR, SAS, MDE and RBU), sectoral (PIN, TRN and RES) and global (GLO) anthropogenic emission reductions. The diagonal, showing the effect
of each region on itself, is underlined. All numbers are rounded to the nearest hundredth, and are shown with standard deviations among models.

| Source regions/sectors | Receptor region | | | | | | |
|---|---|---|---|---|---|---|---|
| | NAM | EUR | SAS | EAS | MDE | RBU | World |
| NAM | -1.33±0.66 | -0.03±0.02 | 0.00±0.01 | -0.02±0.02 | -0.01±0.01 | -0.01±0.01 | -0.08±0.04 |
| EUR | -0.01±0.00 | -1.17±0.87 | -0.01±0.01 | -0.02±0.01 | -0.18±0.08 | -0.26±0.19 | -0.13±0.09 |
| SAS | <-0.01 | <-0.01 | -4.86±2.17 | -0.08±0.08 | -0.03±0.02 | <-0.01 | -1.16±0.51 |
| EAS | -0.03±0.01 | -0.02±0.01 | -0.08±0.07 | -6.19±3.08 | <-0.01 | -0.04±0.02 | -1.45±0.71 |
| MDE | <-0.01 | -0.03±0.01 | -0.12±0.06 | -0.01±0.02 | -0.91±0.38 | -0.05±0.03 | -0.08±0.03 |
| RBU | <-0.01 | -0.07±0.05 | -0.01±0.02 | -0.04±0.02 | -0.03±0.02 | -0.78±0.50 | -0.05±0.03 |
| PIN | -0.61±0.18 | -0.57±0.26 | -1.73±0.71 | -2.75±0.99 | -0.92±0.14 | -0.58±0.19 | -1.46±0.56 |
| TRN | -0.27±0.20 | -0.38±0.41 | -0.82±0.88 | -0.54±0.43 | -0.09±0.06 | -0.15±0.16 | -0.40±0.37 |
| RES | -0.20±0.05 | -0.27±0.12 | -1.93±0.40 | -1.70±0.28 | -0.08±0.02 | -0.20±0.05 | -1.17±0.31 |
| GLO | -1.47±0.72 | -1.52±1.04 | -5.40±2.31 | -6.76±3.29 | -1.55±0.75 | -1.19±0.73 | -3.49±1.51 |


Table 4. Annual multi-model empirical mean $O_3$- and $PM_{2.5}$-related premature deaths with 95% CI from Monte-Carlo simulations in parenthesis
(including uncertainty in baseline mortality rates, RRs and air pollutant concentration across models) in year 2010 baseline. All numbers are
rounded to three significant figures or the nearest 100 deaths. Empirical mean is the mean of 1,000 Monte Carlo simulations.

| | Receptor region | | | | | | |
|---|---|---|---|---|---|---|---|
| | NAM | EUR | SAS | EAS | MDE | RBU | World |
| $O_3$ (11 models) | 15,000 (900 − 30,000) | 13,000 (600 − 28,000) | 136,000 (23,000 − 277,000) | 100,000 (3,900 − 213,000) | 3,200 (300 − 7,000) | 2,900 (100 − 6,600) | 291,000 (30,000 − 596,000) |
| $PM_{2.5}$ (8 models) | 72,000 (1,500 − 158,000) | 203,000 (2,700 − 463,000) | 732,000 (328,000 − 1,110,000) | 1,120,000 (159,000 − 1,720,000) | 79,000 (600 − 133,000) | 177,000 (2,700 − 358,000) | 2,770,000 (514,000 − 4,640,000) |


Table 5. Annual avoided multi-model empirical mean O₃-related premature respiratory deaths with 95% CI from Monte-Carlo simulations in

parenthesis due to 20 % regional (NAM, EUR, SAS, MDE and RBU), sectoral (PIN, TRN and RES) and global (GLO) anthropogenic emission
reductions in each region and worldwide. The diagonal, showing the effect of each region on itself, is underlined. For regional reductions, we also
the RERER (eq. 4) as the percent of total avoided deaths in each receptor region that result from foreign emission reductions, as well as the percent
of global avoided deaths from emission reductions in each source region. All numbers are rounded to three significant figures or the nearest 10
deaths.

| Source regions/sectors | Receptor region | | | | | | | Impact on foreign receptor regions |
|---|---|---|---|---|---|---|---|---|
| | NAM | EUR | SAS | EAS | MDE | RBU | World | |
| NAM | 1,500<br>(-170−4,000) | 330<br>(10−780) | 170<br>(-250−690) | 500<br>(-910−2,200) | 30<br>(0−80) | 70<br>(0−170) | 2,800<br>(-1,300−8,400) | 46% |
| EUR | 60<br>(-80−240) | 930<br>(-70−2,400) | -80<br>(-880−670) | 490<br>(-1,100−2,300) | 50<br>(10−110) | 110<br>(10−250) | 1,700<br>(-490−4,900) | 45% |
| SAS | 40<br>(-40−130) | 50<br>(-30−160) | 19,000<br>(4,000−42,000) | 420<br>(-340−1,400) | 20<br>(0−40) | 10<br>(-10−40) | 20,000<br>(3,600−42,200) | 5% |
| EAS | 230<br>(-50−630) | 310<br>(-50−850) | 450<br>(-1,300−2,400) | 9,700<br>(-2,000−26,400) | 30<br>(0−100) | 80<br>(-10−230) | 11,400<br>(-3,300−31,800) | 15% |
| MDE | 30<br>(-30−120) | 60<br>(-50−190) | 310<br>(-90−910) | 160<br>(-120−520) | 180<br>(-10−480) | 30<br>(0−70) | 870<br>(-330−2,600) | 79% |
| RBU | 40<br>(-60−170) | 150<br>(-50−440) | -200<br>(-1,700−1,200) | 420<br>(-620−1,700) | 20<br>(-10−60) | 140<br>(-60−420) | 640<br>(120−1,300) | 78% |
| PIN | 900<br>(100−2,100) | 850<br>(40−2,100) | 7,400<br>(1,800−15,400) | 7,800<br>(3,100−20,900) | 140<br>(30−330) | 210<br>(-100−650) | 19,300<br>(1,400−45,000) | - |
| TRN | 1,000<br>(-20−2,600) | 970<br>(-270−2,800) | 10,600<br>(2,600−22,000) | 3,500<br>(-420−9,300) | 210<br>(50−440) | 200<br>(20−490) | 18,800<br>(3,000−41,600) | - |
| RES | 200<br>(-20−510) | 250<br>(40−550) | 6,000<br>(1,600−12,200) | 3,000<br>(670−6,300) | 30<br>(0−80) | 60<br>(10−120) | 10,400<br>(2,700−21,100) | - |
| GLO | 2,300<br>(80−5,600) | 2,400<br>(250−5,400) | 22,600<br>(6,200−46,000) | 13,500<br>(1,500−30,300) | 400<br>(80−940) | 550<br>(80−1,210) | 47,400<br>(11,300−99,000) | - |
| RERER | 34% | 61% | 16% | 28% | 55% | 75% | - | |

Table 6. Annual avoided multi-model empirical mean PM$_{2.5}$-related premature deaths (IHD+STROKE+COPD+LC) with 95% CI from Monte-Carlo simulations in parenthesis due to 20 % regional (NAM, EUR, SAS, MDE and RBU), sectoral (PIN, TRN and RES) and global (GLO) anthropogenic emission reductions in each region and worldwide. The diagonal, showing the effect of each region on itself, is underlined. For regional reductions, we also the RERER (eq. 4) as the percent of total avoided deaths in each receptor region that result from foreign emission reductions, as well as the percent of global avoided deaths from emission reductions in each source region. All numbers are rounded to three significant figures or the nearest 10 deaths.

| Source regions/sectors | Receptor region | | | | | | | Impact on foreign receptor regions |
| --- | --- | --- | --- | --- | --- | --- | --- | --- |
| | NAM | EUR | SAS | EAS | MDE | RBU | World | |
| NAM | 18,000 (630−28,300) | 640 (80−1,100) | 10 (-210−80) | 200 (-300−370) | 10 (0−30) | 250 (90−420) | 19,400 (310−30,600) | 7% |
| EUR | 60 (20−110) | 31,900 (4,500−53,900) | 120 (-60−190) | 390 (-20−550) | 400 (30−1,400) | 2,700 (680−8,000) | 39,400 (5,500−63,400) | 19% |
| SAS | 50 (-10−90) | 110 (0−200) | 47,900 (30,000−68,500) | 1,400 (-70−2,100) | 40 (0−150) | 40 (10−110) | 51,300 (32,300−73,300) | 7% |
| EAS | 340 (40−510) | 400 (20−690) | 900 (590−1,400) | 91,100 (440−128,700) | 10 (0−30) | 800 (0−1,300) | 96,600 (3,500−136,000) | 6% |
| MDE | 30 (0−60) | 420 (90−850) | 1,400 (740−2,400) | 180 (-610−460) | 1,600 (240−4,500) | 640 (30−1,600) | 5,000 (1,900−11,100) | 68% |
| RBU | 40 (10−60) | 2,200 (300−3,700) | 90 (-220−190) | 810 (330−1,100) | 80 (10−220) | 17,600 (390−25,700) | 21,500 (900−31,000) | 18% |
| PIN | 9,300 (940−13,000) | 15,700 (1,900−24,700) | 21,000 (8,400−30,700) | 47,310 (22,600−69,700) | 2,200 (200−6,100) | 14,300 (0−24,100) | 128,000 (41,600−179,000) | - |
| TRN | 3,600 (-320−7,000) | 8,900 (130−17,400) | 6,200 (-12,800−14,400) | 6,800 (-6,400−12,200) | 230 (10−770) | 3,100 (0−5,400) | 31,900 (-16,500−58,300) | - |
| RES | 2,900 (110−4,400) | 6,900 (210−11,300) | 25,000 (15,100−40,700) | 29,300 (13,200−52,900) | 200 (10−520) | 4,600 (0−8,100) | 83,400 (41,700−120,000) | - |
| GLO | 19,900 (710−31,300) | 40,900 (4,900−68,100) | 55,300 (36,500−78,300) | 105,000 (4,000−147,000) | 2,800 (330−8,400) | 26,700 (2,300−36,000) | 290,000 (67,100−405,000) | - |
| RERER | 10% | 22% | 13% | 13% | 42% | 34% | - | |


Table 7. Comparison of $O_3$ and $PM_{2.5}$-related premature deaths attributable to PIN,
TRN and RES emissions with previous studies. Results from this study (for 20%
reductions) are multiplied by 5. For Silva et al. (2016), we combine results for "Energy"
and "Industry" to represent PIN, and use "Land transportation" to represent TRN and
"Residential & Commercial" to represent RES. For Lelieveld et al. (2015), we combine
the "Power generation" and "Industry" sectors to represent PIN, and use "Land Traffic"
to represent TRN, and "Residential Energy" to represent RES.

| Emission source sector | This study | Silva et al. (2016) | Lelieveld et al. (2015) |
|---|---|---|---|
| PIN | $O_3$: 96,500 (7,000, 225,000)<br>$PM_{2.5}$: 640,000 (208,000, 895,000) | $O_3$ : 111,000 (23,200, 240,000)<br>$PM_{2.5}$:613,000 (422,000, 816,000) | $O_3 + PM_{2.5}$<br>(692,000) |
| TRN | $O_3$: 94,000 (15,000, 208,000)<br>$PM_{2.5}$: 160,000 (-82,500, 292,000) | $O_3$: 80,900 (17,400, 180,000)<br>$PM_{2.5}$: 212,000 (114,000, 292,000) | $O_3 + PM_{2.5}$<br>(165,000) |
| RES | $O_3$: 52,000 (13,500, 106,000)<br>$PM_{2.5}$:417,000 (209,000, 600,000) | $O_3$: 53,700(12,300, 116,000)<br>$PM_{2.5}$:675,000 (428,000, 899,000) | $O_3 + PM_{2.5}$<br>(1,020,000) |


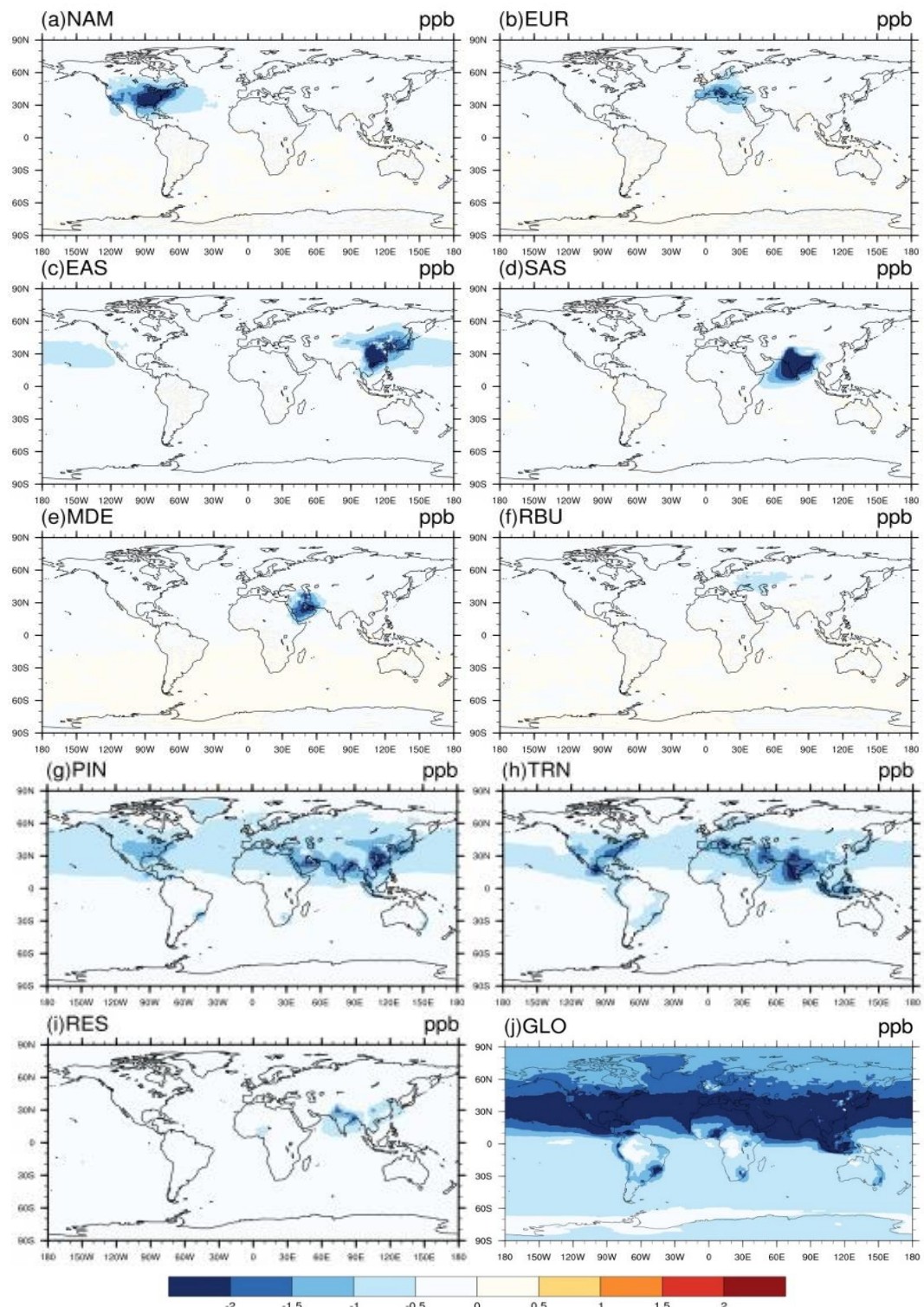


Figure 1– Global difference in multi-model mean O$_3$ concentrations (ppb) in 20% emission reduction scenarios relative to the baseline for the year 2010 in a) North America (NAM), b) Europe (EUR), c) East Asia (EAS), d) South Asia (SAS), e) Middle East (MDE), f) Russia/Belarus/Ukraine (RBU), g) Power and Industry (PIN), h) Transportation (TRN), i) Residential (RES) and j) Global (GLO), shown for the 6-mo. O$_3$ season average of 1-hr. daily maximum health relevant metric.

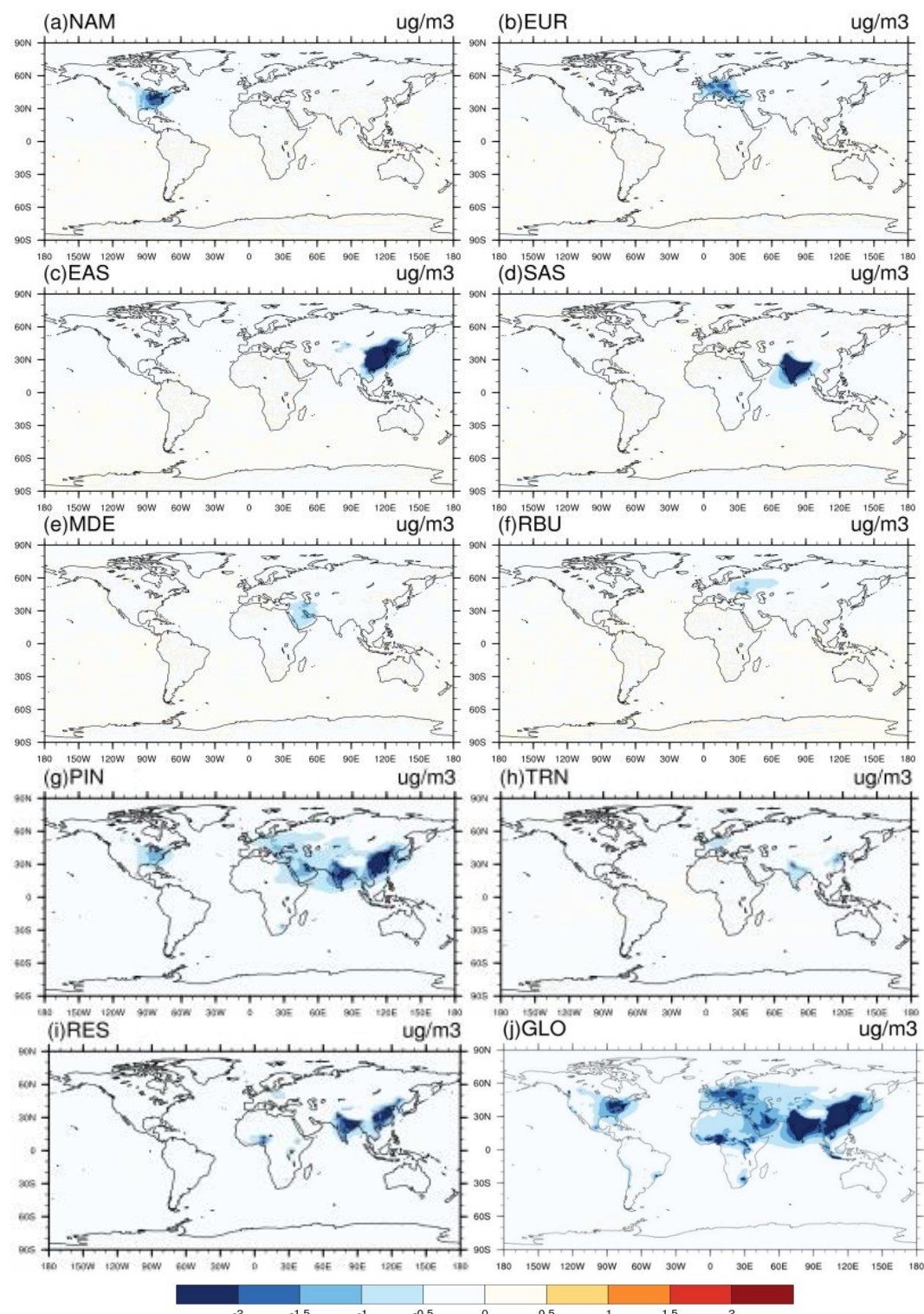


Figure 2– Global difference in multi-model annual mean PM$_{2.5}$ concentrations (μg/m$^3$)
in 20% emission reduction scenarios relative to the baseline for the year 2010 in a)
North America (NAM), b) Europe (EUR), c) East Asia (EAS), d) South Asia (SAS), e)
Middle East (MDE), f) Russia/Belarus/Ukraine (RBU), g) Power and Industry (PIN),
h) Transportation (TRN), Residential (RES) and j) Global (GLO).

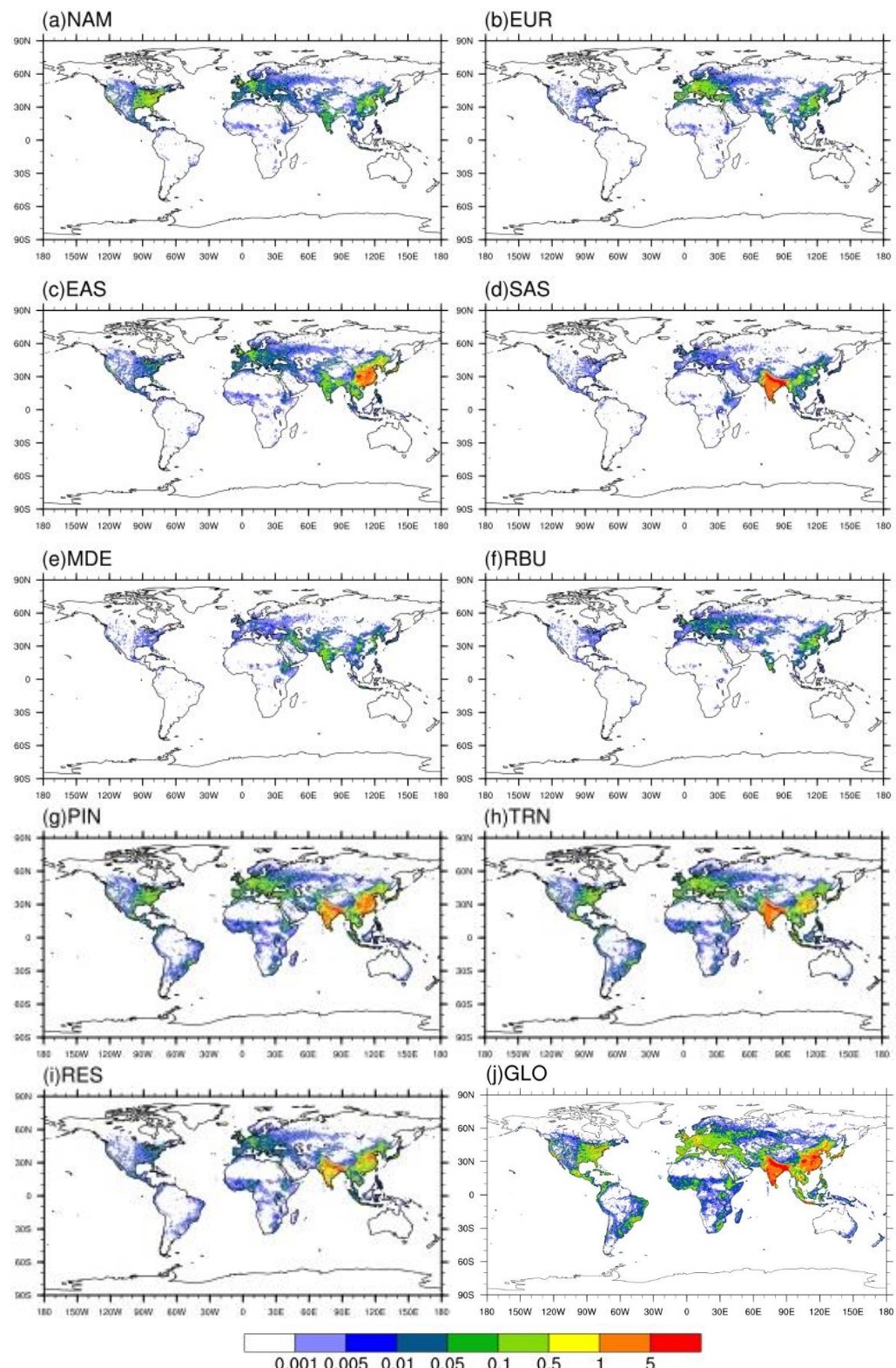

Figure 3. Annual avoided O$_3$-related premature deaths in 2010 per 1,000 km$^2$ due to 20
% emission reduction scenarios relative to the base case in a) North America (NAM),
b) Europe (EUR), c) East Asia (EAS), d) South Asia (SAS), e) Middle East (MDE), f)
Russia/Belarus/Ukraine (RBU), g) Power and Industry (PIN), h) Transportation (TRN),
i) Residential (RES) and j) Global (GLO).

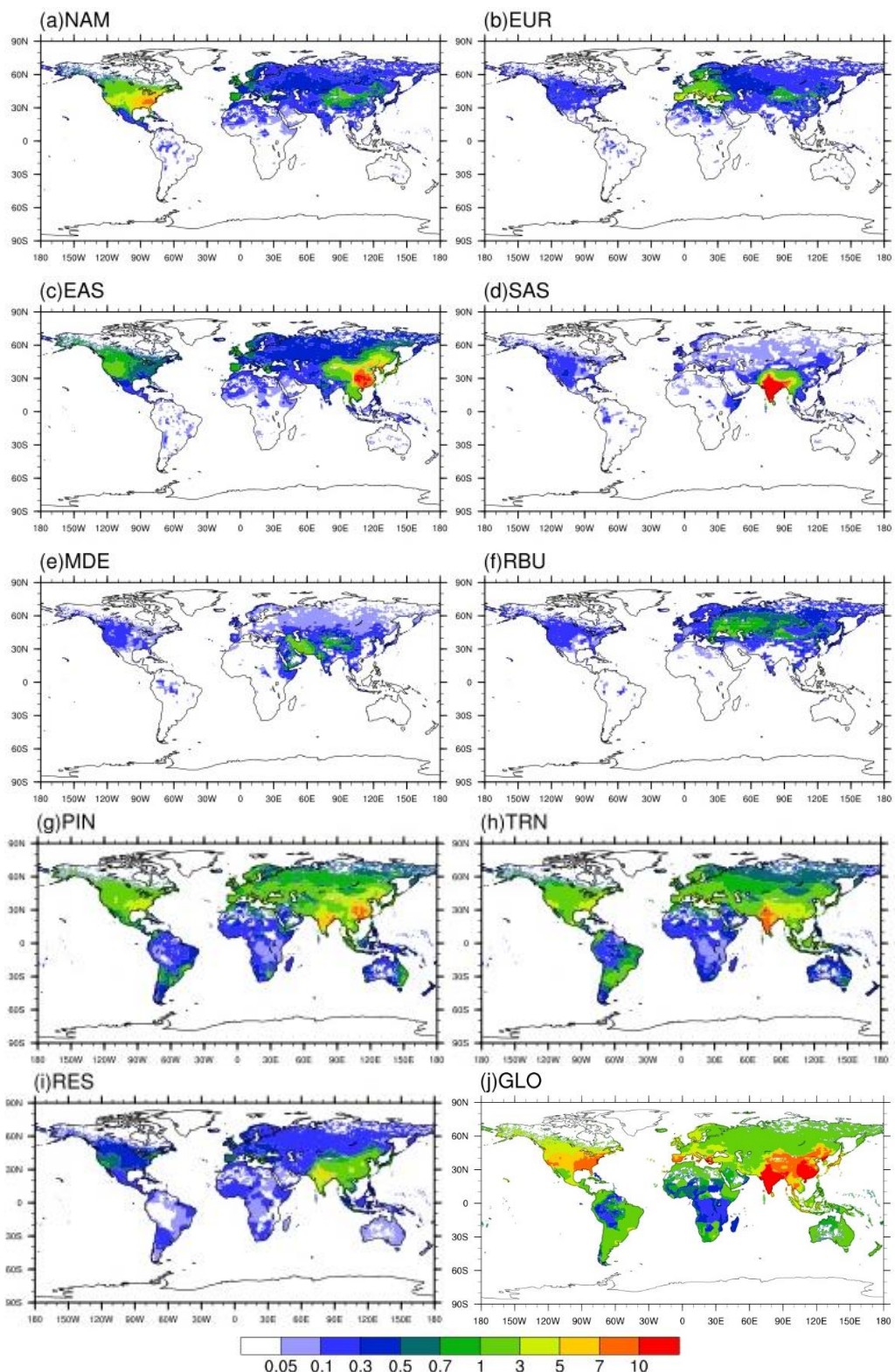

Figure 4. Annual avoided $O_3$-related premature deaths in 2010 per million people due
to 20 % emission reduction scenarios relative to the base case in a) North America
(NAM), b) Europe (EUR), c) East Asia (EAS), d) South Asia (SAS), e) Middle East
(MDE), f) Russia/Belarus/Ukraine (RBU), g) Power and Industry (PIN), h)
Transportation (TRN), i) Residential (RES) and j) Global (GLO)

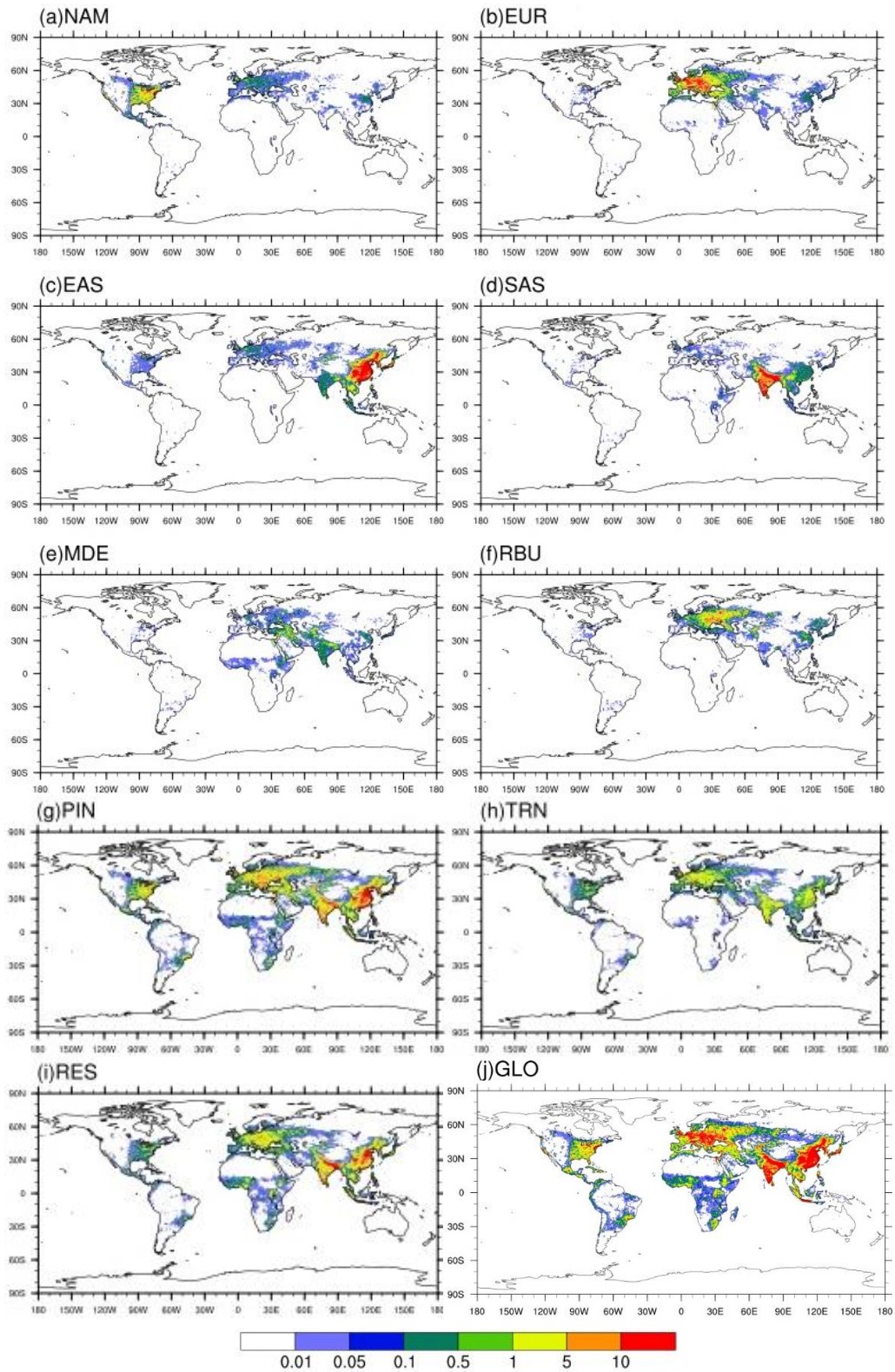

Figure 5. Annual avoided $PM_{2.5}$-related premature deaths in 2010 per 1,000 km$^2$ due to
20 % emission reduction scenarios relative to the base case in a) North America (NAM),
b) Europe (EUR), c) East Asia (EAS), d) South Asia (SAS), e) Middle East (MDE), f)
Russia/Belarus/Ukraine (RBU), g) Power and Industry (PIN), h) Transportation (TRN),
i) Residential (RES) and j) Global (GLO).

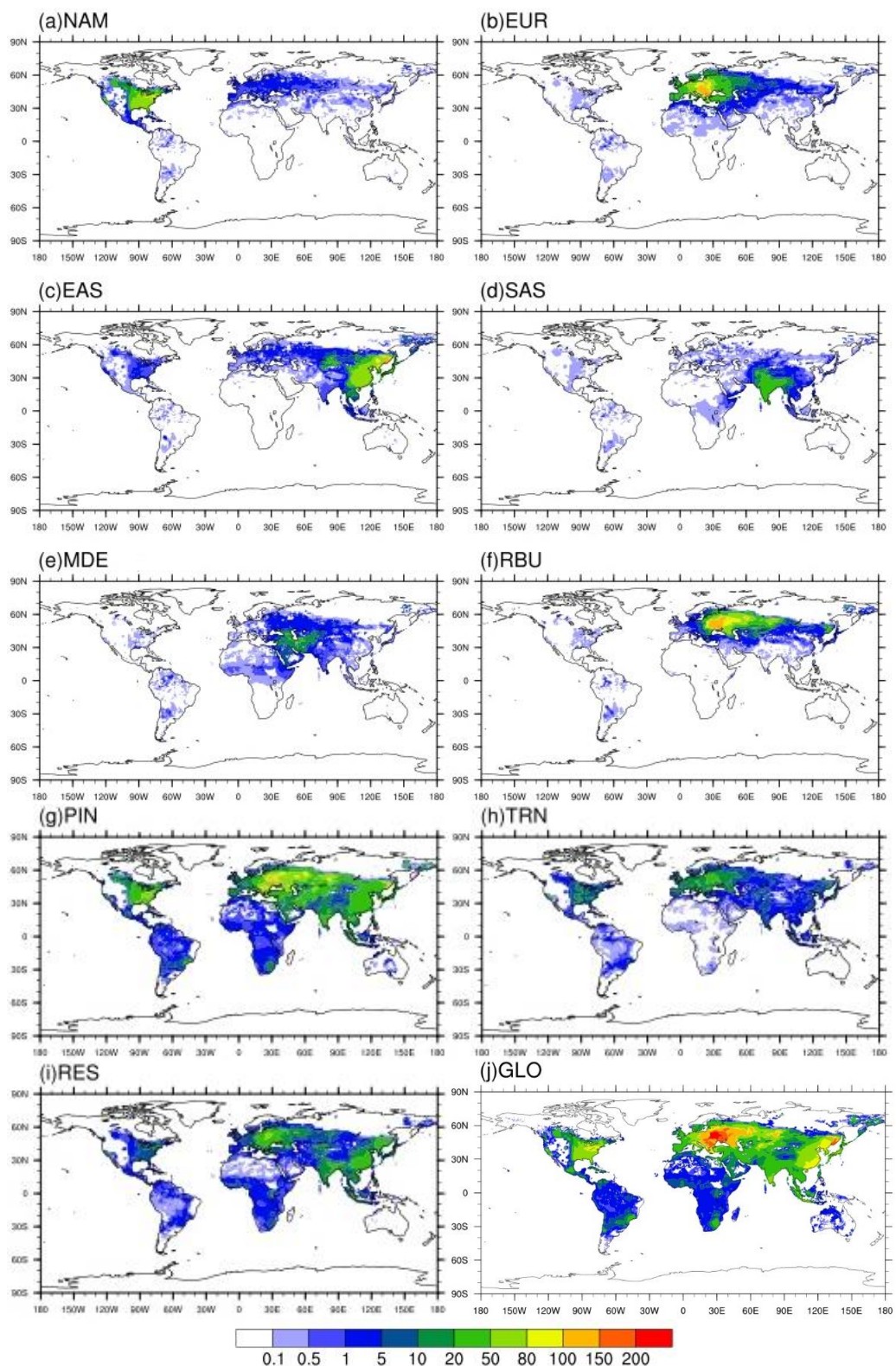

Figure 6. Annual avoided PM$_{2.5}$-related premature deaths in 2010 per million people
due to 20 % emission reduction scenarios) relative to the base case in a) North America
(NAM), b) Europe (EUR), c) East Asia (EAS), d) South Asia (SAS), e) Middle East
(MDE), f) Russia/Belarus/Ukraine (RBU), g) Power and Industry (PIN), h)
Transportation (TRN), i) Residential (RES) and j) Global (GLO).

