# Peer review of "HTAP2 multi-model estimates of premature human mortality due to intercontinental transport of air pollution"

_Atmospheric Chemistry and Physics, 2017_

## Referee Comment (RC1) · Anonymous Referee #1 · 3 Mar 2018

General Comments

This manuscript uses the HTAP2 models to quantify source-receptor relationships for surface ozone and fine particulate matter for emission reductions occurring in six world regions and globally, as well as within three emission sectors. These source-receptor relationships are then combined with concentration-response functions to estimate premature mortality due to intercontinental, within-region, and global emissions (which includes for three separate sectors). This manuscript builds on an existing body of literature coming out of HTAP1, and so, while not particularly novel in terms of methodology, it provides an important benchmark for comparison with earlier and future work.

[Figure]

A serious weakness in the paper is the absence of model comparison to observations. At the very least the paper should include a summary of any evaluation of the HTAP2 models that may be appearing in other articles in this special issue, preferably ones that are already published. A stronger paper would evaluate the specific exposure metrics used to calculate health impacts. For example, observational estimates could be added to Table 1 for regions with ground-level networks. This seems especially relevant in light of the large discrepancies across the HTAP2 models. If some models could be discarded as unrealistic, it is possible that the uncertainty in the estimated numbers of premature mortalities due to the inter-model range may decrease.

In the abstract, some context could be provided as to whether the numbers here are in line with earlier work.

Specific Comments

Lines 63-68. Does this mean outside of any of the six regions?

Many prior studies are mentioned in the introduction. Are there any robust findings across this prior body of work?

Lines 246-248. Is the actual value of $\beta$ given somewhere?

Line 261. Make sure all terms in equation 3 are defined.

Line 267-268. Elaborate on Zcf: does it vary from 5.8 to 8.8 $\mu$g m-3 in space and time?

Figures S8 and S9 are referred to several times in the text but are impossible to read. I suggest splitting them each into 4 figures, with half the models on each, one for the regional perturbations and one for the sectoral perturbations. The full range of the colorbar isn't used, so consider using a different color bar that allows for one to read the values off the figure more easily.

Lines 318-320. Is this intended to be a quantitative comparison? If so, are the metrics reported here and in the Lin et al. studies the same?

[Figure]

Lines 449-459. This seems like methodology and could be included in the earlier section.

Lines 545-547. Could the use of a different year make a difference here?

Lines 559-560. This seems like an important point and suggest including in abstract and conclusions.

Lines 609-610. Given the large ranges, is it really meaningful to report averages?

Table 4. What is an "empirical mean"?

Table S1. Why not calculate PM2.5 consistently across models from the individual components?

---

## Referee Comment (RC2) · Anonymous Referee #2 · 14 Mar 2018

This manuscript uses the multi-model results from HTAP2 project to estimate mortality for the baseline year 2010, and health benefits from reduced emissions in source regions. In general, it is well organized and written, and the multi-model results can provide more reasonable range than single model results in previous studies. However, some details are not well documented and explanations are too general, but important for readers.

Specific Comments:

1. Page 3, line 100: It is better to provide some brief explanation of reasons for large differneces in estimates (4.2 and 2.1 million premature deaths)

[Figure]

2. Page 5, line 159: Please specify if the perturbation is increasing or decreasing.

3. Page 6, line 190-203: how do these models perform in simulating ozone and PM2.5

4. Page 7, line 246-257: what beta value is used in this study? any source for the used RR=1.040? Please clarify.

5. Page 8, line 264-271: The used RR framework here is not acturally the latest. Please refer to Cohen et al. (2017).

6. Page 8, line 276-277: Please clarify how you treat age distribution in the 2011 populaiton dataset.

7. Page 8, is sex difference considered in the estimation?

8. Page 8, line 282: Monte Carlo simulation is powerful to address uncertainty issues. However, the way of including model air pollutant concentrations is a bit misleading. The procedure in this study is actually the range of multi-model results. However, it is possible that this range deviate from the observations. Without showing model evaluation, we don't have confidence how reliable is the range from multi-models.

9 Page 9, line 306: The texts refer to supplemental plots many times. I would suggest move some important figures from supplemental materials.

10 Page 10, line 368-369: Please provide more details here: the updated baseline mortality rate in 2017, and how population is different. This comparison is too general here. In my understanding, the biggest change from GBD framework from old to latest (Cohen et al., 2017) is not just baseline mortality. In Cohen et al. (2017), the RR for stoke is totally different from previous version GBD, and LRI disease is added in addition to IHD, LC, COPD and stroke.

11 Page 11 line 382-383: Please clarify how the avoid deaths is calculated. the IER model is not linear: at the high end large changes in polluant will not reulst in large changes in death, some studies used average changes, some used marginal. How is

this addressed here?

12 Page 11 line 406-408: The explanation here is not convincing.

13 It would be great to make a table to inter-compare the response of sector reductions, which is highly uncertain from different models, and please discuss it too.

---

## Author Comment (AC1) · 16 May 2018

**Response to Referee #1:**

*General comments*

*This manuscript uses the HTAP2 models to quantify source-receptor relationships for surface ozone and fine particulate matter for emission reductions occurring in six world regions and globally, as well as within three emission sectors. These source-receptor relationships are then combined with concentration-response functions to estimate premature mortality due to intercontinental, within-region, and global emissions (which includes for three separate sectors). This manuscript builds on an existing body of literature coming out of HTAP1, and so, while not particularly novel in terms of methodology, it provides an important benchmark for comparison with earlier and future work.*

Thank you for your careful review of our paper and constructive comments.

*A serious weakness in the paper is the absence of model comparison to observations. At the very least the paper should include a summary of any evaluation of the HTAP2 models that may be appearing in other articles in this special issue, preferably ones that are already published. A stronger paper would evaluate the specific exposure metrics used to calculate health impacts. For example, observational estimates could be added to Table 1 for regions with ground-level networks. This seems especially relevant in light of the large discrepancies across the HTAP2 models. If some models could be discarded as unrealistic, it is possible that the uncertainty in the estimated numbers of premature mortalities due to the inter-model range may decrease.*

**Response:**

Thank you for this comment. We had previously anticipated that other HTAP2 studies would include this comparison with observations. But we now see that while two papers do include comparisons in some regions, a full global comparison with observations for all of the models used in this study is desirable here. We have now included this model evaluation with ground level observations as described in the new section 2.2 (Lines 241-294):

[revised manuscript text omitted]

*In the abstract, some context could be provided as to whether the numbers here are in line with earlier work.*
**Response:**
For impacts of intercontinental transport, we compare results from TF-HTAP2 with the previous TF-HTAP (Anenberg et al., 2009; 2014) and comparable studies (West et al., 2009; Duncan et al., 2008), and for sectoral reductions, we compare with previous studies by Crippa et al (2017), Lelieveld et al. (2015) and Silva et al. (2016a). We have modified the abstract to compare regional results with previous studies (Lines 63-66):
"Our findings that most avoided $O_3$-related deaths from emission reductions in NAM and EUR occur outside of those regions contrast with those of previous studies, while estimates of $PM_{2.5}$-related deaths from NAM, EUR, SAS and EAS emission reductions agree well."

And we have also modified the abstract to compare sectoral impacts (Lines 75-81):
"In sectoral emission reductions, TRN emissions account for the greatest fraction (26-53% of global emission reduction) of $O_3$-related premature deaths in most regions, in agreement with previous studies, except for EAS (58%) and RBU (38%) where PIN emissions dominate. In contrast, PIN emission reductions have the greatest fraction (38-78% of global emission reduction) of $PM_{2.5}$-related deaths in most regions, except for SAS (45%) where RES emission dominates, which differs with previous studies in which RES emissions dominate global health impacts."

***Specific comments***
*1.Lines 63-68. Does this mean outside of any of the six regions?*
**Response:**
This metric was not sufficiently clear in the previous draft. We now present two estimates of the impact of intercontinental transport on mortality, from the source and receptor perspectives, which are added to Tables 5 and 6. The estimate of the impact of intercontinental transport on mortality from the receptor perspective uses

the RERER metric that was introduced in previous HTAP studies. We express extra-regional deaths, as presented in the abstract, as the total avoided deaths outside of each source regions from six source emission reductions. We modified how this is presented in the abstract (Lines 67-70):

"For six regional emission reductions, the total avoided extra-regional mortality is estimated as 6,000 (-3,400, 15,500) deaths/year and 25,100 (8,200, 35,800) deaths/year through changes in $O_3$ and $PM_{2.5}$, respectively."

We added text to clarify how the RERER metric is defined (Lines 384-396):

"We estimate the impacts of extra-regional emission reductions on mortality by using the Response to Extra-Regional Emission Reduction (RERER) metric defined by TF-HTAP (Galmarini et al., 2017):

$$RERER_i = \frac{R_{global} - R_{region,i}}{R_{global}} \quad (4)$$

where for a given region $i$, $R_{global}$ is the change in mortality in the global 20% reduction simulation (GLO) relative to the base simulation, and $R_{region,i}$ is the change in mortality in response to the 20% emission reduction from that same region $i$. A RERER value near 1 indicates a strong relative influence of foreign emissions on mortality within a region, while a value near 0 indicates a weak foreign influence. We also estimate the total avoided extra-regional mortality from a source perspective as the sum of avoided deaths outside of each of the 6 source regions, and from a receptor perspective by summing $R_{global} - R_{region,i}$ for all 6 regions."

We modified how the results are presented on these issues (Lines 558-561):
"Overall, adding results from all 6 regional reductions, interregional transport of air pollution from extra-regional contributions is estimated to lead to more avoided deaths through changes in $PM_{2.5}$ (25,100 (8,200, 35,800) deaths/year) than in $O_3$ (6,000 (-3,400, 15,500) deaths/year), consistent with Anenberg et al. (2009; 2014)."

We modified this in the discussion (Lines 649-653):
"Also, total avoided deaths through interregional air pollution transport are estimated as 6,000 (-3,400, 15,500) deaths/year for $O_3$ and 25,100 (8,200, 35,800) deaths/year for $PM_{2.5}$ in this study, in contrast with 7,300 (3,600, 11,200) deaths/year for $O_3$ and 11,500 (8,800, 14,200) deaths/year for $PM_{2.5}$ in Anenberg et al. (2009; 2014)."

And we modified this in the conclusions (Lines 732-735):
"Reductions from all six regions in the transport of air pollution between regions are

estimated to lead to more avoided deaths through changes in PM$_{2.5}$ (25,100 (8,200, 35,800) deaths/year) than for O$_3$ (6,000 (-3,400, 15,500) deaths/year)."

*2.Many prior studies are mentioned in the introduction. Are there any robust findings across this prior body of work?*

**Response:**

We have modified the introduction to point out the robust findings by prior studies (Lines 140-143):

"These prior studies have consistently concluded that most avoided O$_3$-related deaths from emission reductions in NAM and EUR occur outside of those regions, while most avoided PM$_{2.5}$-related deaths occur within the regions."

*3.Lines 246-248. Is the actual value of $\beta$ given somewhere?*

**Response:**

We have added text to give the value of RR from the Jerrett study, from which Beta is calculated from equation 1 (Lines 316-318):

"For O$_3$, RR = 1.040 (95% Confidence Interval, CI: 1.013-1.067) for a 10 ppb increase in O$_3$ concentrations (Jerrett et al., 2009), which from eq. 1 gives values for $\beta$ of 0.00392 (0.00129-0.00649)."

*4.Line 261. Make sure all terms in equation 3 are defined.*

**Response:**

Burnett et al. (2014) defines the function given and specifies three parameters (α, γ, δ) which they use to allow more flexibility in fitting the cause-specific RR. These parameters therefore do not have specific physical meaning, and are used in the functional fitting, so we refer the reader to Burnett's paper to understand these parameters more fully (Lines 329-334):

"RR is calculated as:

For $z < z_{cf}$, $RR_{IER}(z) = 1$   (2)

For $z \geq z_{cf}$, $RR_{IER}(z) = 1 + \alpha\{1 - exp[-\gamma(z - z_{cf})^\delta]\}$   (3)

where z is the PM$_{2.5}$ concentration in μg/m$^3$ and $z_{cf}$ is the counterfactual concentration below which no additional risk is assumed, and the parameters α, γ, and δ are used to fit the function for cause-specific RR (Burnett et al., 2014)."

*5.Line 267-268. Elaborate on Zcf: does it vary from 5.8 to 8.8 g/m$^3$ in space and time?*

**Response:**

We have revised to clarify (Lines 338-341):

"A uniform distribution from 5.8 μg/m$^3$ to 8.8 μg/m$^3$ is used for $z_{cf}$ as suggested by

Burnett et al. (2014), which does not vary in space nor time. For uncertainty analysis, we use results from 1,000 Monte Carlo simulations of Burnett et al. (2014) to calculate RR in each grid cell by eq.2 or eq. 3."

*6.Figures S8 and S9 are referred to several times in the text but are impossible to read. I suggest splitting them each into 4 figures, with half the models on each, one for the regional perturbations and one for the sectoral perturbations. The full range of the colorbar isn't used, so consider using a different color bar that allows for one to read the values off the figure more easily.*

**Response:**

We split Figures S8-S9 into 6 pages, and we use a different color bar to show full range of data. See the updated Figs. S14-S17.

*7.Lines 318-320. Is this intended to be a quantitative comparison? If so, are the metrics reported here and in the Lin et al. studies the same?*

**Response:**

No, we don't intend to show a quantitative comparison with Lin et al. (2012 and 2017) due to the different ozone metrics evaluated. Instead, we suggest that the ozone responses in the western US to emission reductions from EAS are similar to those of Lin et al. (2012 and 2017) who show that a model can capture the measured western US ozone increases due to rising Asian emissions. We add this text (Lines 423-426):

"Our ensemble shows similar ozone responses in the western US to emission reductions from EAS (Figs. 1c) as those modeled by Lin et al. (2012 and 2017), who show that a model can capture the measured western US ozone increases due to rising Asian emissions."

*8.Lines 449-459. This seems like methodology and could be included in the earlier section.*

**Response:**

We have moved these lines into the method section (Lines 376-382):

"We also quantify the uncertainties in mortality due to the spread of air pollutant concentrations across models, RRs, and baseline mortality rates, as contributors to the overall uncertainty, expressed as a coefficient, of variation and compare the result with the Monte-Carlo analysis estimate. To do so, we hold two variables at their mean values and change the variable of interest within its uncertainty range; for example, using mean RRs and baseline mortality rates, we analyze the spread of the model ensemble to calculate the coefficient of variation caused by model

uncertainty."

9.Lines 545-547. *Could the use of a different year make a difference here?*
**Response:**
We agree with the reviewer that the different year could be responsible for part of the differences between studies. We have revised the text (Lines 653-657):
"These differences likely result from different concentration-response functions and the use of 6 regions here vs. 4 by Anenberg et al. (2009; 2014). In addition, updated atmospheric models and emissions inputs, as well as different atmospheric dynamics in the single years chosen in TF-HTAP1 vs. TF-HTAP2 may contribute to the differences."

10.Lines 559-560. *This seems like an important point and suggest including in abstract and conclusions.*
**Response:**
We have revised the abstract to add this comment (Lines 72-75):
"For NAM and EUR, our estimates of avoided mortality from regional and extra-regional emission reductions are comparable to those estimated by regional models for these same experiments."

And we have added it to the conclusions (Lines 735-738):
"For NAM and EUR, our estimates of avoided mortality from regional and extra-regional emission reductions are comparable to those estimated by regional models in AQMEII3 (Im et al., 2018) for these same emission reduction experiments."

11.Lines 609-610. *Given the large ranges, is it really meaningful to report averages?*
**Response:**
The overall percentage is derived from all 6 regional emission reductions altogether, not the average of percentages for each region. We've revised to clarify (Lines 722-727):
"For regional scenarios, 6 source emission reductions altogether can cause 84% of the global avoided $O_3$-related premature deaths within the source region, ranging from 21 to 95% among 6 regions, and 16% (5 to 79%) outside of the source region. For $PM_{2.5}$, 89% of global avoided $PM_{2.5}$-related premature deaths are within the source region, ranging from 32 to 94% among 6 regions, and 11% (6 to 68%) outside of the source region."

12.Table 4. *What is an "empirical mean"?*

**Response:**

Since we conduct 1,000 Monte Carlo simulations to propagate uncertainty from baseline mortality rates, modeled air pollutant concentrations, and the RRs in the health impact functions, the mean of the result is called the "empirical mean" as the mean of 1,000 simulations. We added this explanation into Table 4:

"Empirical mean is the mean of 1,000 Monte Carlo simulations."

We also revised the method section to explain where this result is used (Lines 373-375):

"The mean of the 1,000 Monte Carlo simulations (the "empirical mean") may differ from the mean when using the mean RR."

*13.Table S1. Why not calculate PM$_{2.5}$ consistently across models from the individual components?*

**Response:**

Different models use different functions to represent PM$_{2.5}$, that are appropriate for each model based on how different species are defined in the models. We choose to use the reported PM$_{2.5}$ from each model, rather than to recalculate PM$_{2.5}$ based on their reported species concentrations. We include the functions used by each model in Table S1 to communicate the species that each model simulated, and other modeling differences, where for example some models may be missing important species, but we do not apply these functions ourselves in this study.

---

## Author Comment (AC2) · 16 May 2018

**Response to Referee #2**

*General comments*

*This manuscript uses the multi-model results from HTAP2 project to estimate mortality for the baseline year 2010, and health benefits from reduced emissions in source regions. In general, it is well organized and written, and the multi-model results can provide more reasonable range than single model results in previous studies. However, some details are not well documented and explanations are too general, but important for readers.*

Thank you for your careful review of our paper and constructive comments.

*Specific comments*

*1.Page 3, line 100: It is better to provide some brief explanation of reasons for large differneces in estimates (4.2 and 2.1 million premature deaths).*

**Response:**

We have added short discussion on this point (Lines 107-109):

"These differences in GBD estimates result mainly from differences in concentration response functions and estimates of pollutant concentrations."

*2.Page 5, line 159: Please specify if the perturbation is increasing or decreasing.*

**Response:**

We reduced the anthropogenic emissions by 20% (Line 170):

"Anthropogenic emissions were reduced by 20% in six source regions: …"

*3.Page 6, line 190-203: how do these models perform in simulating ozone and PM$_{2.5}$*

**Response:**

Thank you for this comment. We had previously anticipated that other HTAP2 studies would include this comparison with observations. But we now see that while two papers do include comparisons in some regions, a full global comparison with observations for all of the models used in this study is desirable here. We have now included this model evaluation with ground level observations as described in the new section 2.2 (Lines 241-294):

[revised manuscript text omitted]

4.Page 7, line 246-257: what beta value is used in this study? any source for the used RR=1.040? Please clarify..

**Response:**

We added this (Lines 316-318):

"For $O_3$, RR = 1.040 (95% Confidence Interval, CI: 1.013-1.067) for a 10 ppb increase in $O_3$ concentrations (Jerrett et al., 2009), which from eq. 1 gives values for $\beta$ of 0.00392 (0.00129-0.00649)."

5.Page 8, line 264-271: The used RR framework here is not actually the latest. Please refer to Cohen et al. (2017).

**Response:**

Our work was nearly completed before Cohen et al. (2017) was published, and so we chose the most recent available RR from Burnett et al. (2014) for $PM_{2.5}$. This function was widely used in many studies, including by Silva et al (2016a), Lelieveld et al (2015), and GBD 2010 (Lim et al., 2012). However, we have added short discussion on this difference (Lines 473-481):

"Cohen et al. (2017) use RRs for particulate matter for IHD and stroke mortality that are modified from those used by Burnett et al (2014) and applied age modification to the RRs, fitting the IER model for each age group separately. The updated IER with estimated higher relative risks, together with greater global pollution and baseline mortality rates in the low-income and middle-income countries in east and south Asia leads to the higher absolute numbers of attributable deaths and disability-adjusted life-years in GBD 2015 than estimated in GBD 2013 (Forouzanfar et al., 2016). Also, GBD 2015 includes child lower respiratory infections estimate whereas we do not".

6.Page 8, line 276-277: Please clarify how you treat age distribution in the 2011

*populaiton dataset.*

**Response:**

We add text (Lines 355-358):

"For the population of adults aged 25 and older, we use ArcGIS 10.2 geoprocessing tools to estimate the population per 5-year age group in each cell by multiplying the country level percentage in each age group by the population in each cell."

*7.Page 8, is sex difference considered in the estimation?*

**Response:**

No, we only consider age-specific RR, as given by the health impact functions we use and the underlying epidemiological studies.

*8.Page 8, line 282: Monte Carlo simulation is powerful to address uncertainty issues. However, the way of including model air pollutant concentrations is a bit misleading. The procedure in this study is actually the range of multi-model results. However, it is possible that this range deviate from the observations. Without showing model evaluation, we don't have confidence how reliable is the range from multi-models.*

**Response:**

We've added the model evaluation in section 2.2 (Lines 241-294).

We also added an acknowledgement that the range of models in an ensemble is not a true reflection of the uncertainty in emissions to the method section (Lines 371-373):

"One should acknowledge that the range of models in an ensemble is not a true reflection of the uncertainty in emissions to concentration relationships."

*10.Page 9, line 306: The texts refer to supplemental plots many times. I would suggest move some important figures from supplemental materials.*

**Response:**

We've moved figures S6-S7 to figures 1-2 in the main paper. The order of figures has been updated to reflect this change in main paper as well as the supporting document.

*11.Page 10, line 368-369: Please provide more details here: the updated baseline mortality rate in 2017, and how population is different. This comparison is too general here. In my understanding, the biggest change from GBD framework from old to latest (Cohen et al., 2017) is not just baseline mortality. In Cohen et al. (2017), the RR for stoke is totally different from previous version GBD, and LRI disease is added in addition to IHD, LC, COPD and stroke.*

**Response:**

As stated before, we now provide details on how RRs were updated for use by Cohen et al. (2017) (Lines 473-481):

"Cohen et al. (2017) use RRs for particulate matter for IHD and stroke mortality that are modified from those used by Burnett et al (2014) and applied this age modification to the RRs, fitting the IER model for each age group separately. The updated IER with estimated higher relative risks, together with greater global pollution and baseline mortality rates in the low-income and middle-income countries in east and south Asia leads to the higher absolute numbers of attributable deaths and disability-adjusted life-years in GBD 2015 than estimated in GBD 2013 (Forouzanfar et al., 2016). Also, GBD 2015 includes child lower respiratory infections estimate whereas we do not."

*12.Page 11 line 382-383: Please clarify how the avoid deaths is calculated. the IER model is not linear: at the high end large changes in pollutant will not result in large changes in death, some studies used average changes, some used marginal. How is this addressed here?*

**Response:**

The percentage of the global change in $O_3$-related deaths within the source region is computed by the number of avoided deaths within source region divided by the number of avoided deaths globally from 20% source emission reduction. We've revised to clarify this calculation (Lines 495-496):

"The number of avoided deaths within source region is divided by the number of avoided deaths globally"

We added text to discuss the issue about IER model (Lines 343-352):

"However, in the IER model, the concentration–response function flattens off at higher $PM_{2.5}$ concentrations, yielding different estimates of avoided premature mortality for identical changes in air pollutant concentrations from less-polluted vs. highly-polluted regions. That is, one unit reduction of air pollution may have a stronger effect on avoided mortality in regions where pollution levels are lower (e.g., Europe, North America) compared with highly polluted regions (e.g., East Asia, India), which would not be the case for a log-linear function (Jerrett et al., 2009; Krewski et al., 2009). Therefore, using the IER model in this study may result in smaller changes in avoided mortality in highly polluted areas than using the linear model."

*13.Page 11 line 406-408: The explanation here is not convincing.*

**Response:**

We've revised this explanation to (Lines 520-522):

"In addition, updated atmospheric models and emissions inputs, as well as different atmospheric dynamics in the single years chosen in TF-HTAP1 vs. TF-HTAP2 may contribute to the differences."

*14.It would be great to make a table to inter-compare the response of sector reductions, which is highly uncertain from different models, and please discuss it too.*
**Response:**
We've listed this inter-comparison between models for sector reductions in TableS9-S10 and discussed these differences in Lines 616-625:

"Considering results from individual models, we found that $O_3$-and $PM_{2.5}$-related mortality from TRN emission reductions show greater relative uncertainty than from PIN or RES (Table 5-6 and Table S9-S10), reflecting a greater spread of results across models. Regional impacts from individual models also differ from the ensemble mean result - e.g., for $O_3$, GEOSCHEMADJOINT and OsloCTM3.v2 show that reducing PIN emissions causes the greatest fraction of avoided $O_3$-related deaths in EUR, while GEOSCHEMADJOINT, HadGM2-ES and OsloCTM3.v2 show that TRN emissions have the greatest fraction of avoided $O_3$-related deaths in RBU (Figs. S20). For $PM_{2.5}$, CHASER_t42 and GEOSCHEMADJOINT show that reducing PIN emissions causes the greatest fraction of avoided $PM_{2.5}$-related deaths in SAS (Figs. S21)"

In addition, we also compare our $O_3$ and $PM_{2.5}$-related premature deaths attributable to PIN, TRN and RES emissions with previous studies conducted by Silva et al. (2016) and Lelieveld et al. (2015) in table 7 and discuss the differences from our estimates in Lines 601-615 :

"In comparison with other studies (Table 7), our conclusion that PIN emissions cause the most $O_3$-related deaths and TRN emissions cause the greatest fraction of avoided deaths in most regions agrees well with Silva et al (2016a). For $PM_{2.5}$, reducing PIN emissions avoids the most $PM_{2.5}$-related premature deaths globally (128,000 (41,600, 179,000) deaths/year) and in most regions (38-78% of the global emission reduction), except for SAS (45%) where the RES emission dominates. Although these findings differ from those of Lelieveld et al (2015) and Silva et al (2016), who find that Residential emissions have the greatest of impact on $PM_{2.5}$ mortality globally and in most regions, all studies agree that PIN emissions have the greatest impact in NAM. Our result is also comparable with Crippa et al (2017) who find that PIN emissions have the greatest health impact in most countries. Although comparable emission inventories are used (i.e. Lelieveld et al (2015) use EDGAR emissions while Silva et al (2016) use RCP8.5. emissions), our lower mortality estimate for RES emissions may

be explained by our 20% reductions relative to the zero-out method, and the different years simulated."

and Lines 674-686 :

"Differences in our estimates of premature mortality attributable to air pollution from three emission sectors (multiplied by 5) may be explained by methodological differences relative to previous studies (Silva et al., 2016; Lelieveld et al., 2015), including our use of 20% emission reductions versus the zero-out method in those studies, different emission inventories, a multi-model ensemble versus single models, and differences in baseline mortality rates, population, and concentration response functions. Our finding that TRN emissions contribute the most avoided deaths for $O_3$ in most regions agrees well with the result by Silva et al (2016a), but differs for $PM_{2.5}$ mortality for which we find that PIN emissions cause the most deaths, while both Silva et al (2016a) and Lelieveld et al (2015) find that RES emissions are responsible for the most deaths. This discrepancy may be explained by different $PM_{2.5}$ species included in individual models, as we showed that changes in $PM_{2.5}$ concentration to TRN emission differ across models."